

# Refinement of the fluidity parameter range with a stress exponent of four in Glen's law: insights from Antarctic bed topography model

Su-Jeong Lim[1], Yi-Jeong Baek[1], Byung-Dal So[1*]

[1]Department of Geophysics, Kangwon National University, Chuncheon, Republic of Korea

*Correspondence to*: Byung-Dal So (bdso@kangwon.ac.kr)

**Abstract.** Understanding the stress-dependent behaviour of the ice sheet is critical to projecting ice mass balance. Glen's law is used to calculate ice viscosity, conventionally with a stress exponent (n) equal to three. However, a stress exponent of four has recently been proposed for ice dynamics. The suggested range of the fluidity parameter (A) for n = 4 is of the order of six

(i.e., $10^{-35}$ to $10^{-29}$ $Pa^{-4}s^{-1}$), leading to a significant uncertainty in ice velocity than when n = 3. Here, we refined A to within one order, aligning with observed Antarctic ice velocities with simplified slope and Antarctic bed topography models. The Antarctic bed topography models, based on Antarctic BedMap2 data, specifically include the Ronne, Thwaites, and Ross Ice Shelves for West Antarctica, and the Amery, Shackleton Ice Shelves, and Mertz Glacial Tongue for East Antarctica. We found that the simplified model and the West and East Antarctic models share a common range of A values for matching

observed Antarctic ice velocities. Refinement narrows the range of A to within one order for both the simplified and West/East Antarctic models. The narrowed range for A is from $4.0 \times$ to $16.0 \times 10^{-32}$ $Pa^{-4}s^{-1}$. Both models have common A values for representing Antarctic ice velocity, providing an insight into the applicable A values.

## 1 Introduction

Ice mass balance is controlled by rapid variations in ice flow due to changes in basal friction, ice stiffness, and ice

melting that occur around the grounding line between the grounded ice sheet and the floating ice shelf (Rignot et al., 2019; Morlighem et al., 2021; Dawson et al., 2022). Future greenhouse gas emission scenarios calibrated to the last interglacial sea level suggest that Antarctic ice loss may cause a sea level rise of approximately 15 m by 2500 (DeConto and Pollard, 2016). Projection of ice flow requires ice sheet dynamics modeling that simultaneously includes various factors such as bed topography, glacier geometry, basal friction, and hydrostatic pressure, based on the nonlinearity of ice viscosity as a function

of stress (Marshall, 2005; Durand et al., 2011). The velocity field of a gravitation-driven ice flow is strongly influenced by ice viscosity, which constitutes the relationship between the shear strain rate and the stress of the ice. Due to the nonlinearity of ice viscosity concerning stress and temperature, grounding line projection requires ice sheet dynamics modeling, which integrates various factors such as bed topography, glacier geometry, basal friction, and hydrostatic pressure. Glen's law (Glen, 1952; Glen, 1955), where the strain rate is proportional to the fluidity parameter A and the $n^{\text{th}}$ power of the deviatoric stress,

describes the rheological behavior of ice with the shear viscosity. By incorporating geodetic ice creep data (Cuffey and



Kavanaugh, 2011; Behn et al., 2021), ice sheet modeling has suggested an n value of approximately three, which effectively reproduces the surface ice velocity in Antarctica (Martin and Sanderson, 1980; Pattyn et al, 2012), such as the Amery Ice Shelf (Thomas, 1973; Hamley et al., 1985).

However, using a single value for the stress exponent potentially oversimplifies ice rheology (Behn et al., 2021), which is modulated by multiple factors (Goldsby and Kohlstedt, 2001). A detailed depth analysis of deep ice cores from Antarctica and Greenland revealed large variations in grain shape, grain size, and anisotropy within the ice structure (Budd and Jacka, 1989; Cuffey et al., 2000), indicating that stress exponent values other than three can be considered when fitting ice viscosity. Parametric search modeling to determine the value of n for explaining the surface velocity along the flow line of the Vatna Glacier, the largest ice cap (e.g., Iceland; Marshall, 2005) and terminus ice shelves (e.g., Taylor Glacier; Cuffey and
Kavanaugh, 2011) in Antarctica deduced stress exponents n = 2.5–5.

The A is modulated by the temperature, interstitial water content, and hydrostatic pressure (Goldsby and Kohlstedt, 2001, Moore, 2014). The value of A derived from a laboratory ice deformation experiment with n = 3 ranged from 1.8 to $93 \times 10^{-25}$ $Pa^{-3}s^{-1}$ (MacAyeal et al., 1998), which is conventionally accepted in ice sheet dynamics modeling (e.g., Budd and Jacka, 1989; Schoof, 2007; Favier et al., 2012; Pattyn et al., 2013; Hossain et al., 2020; Millstein et al., 2022). Furthermore,
the previous ice sheet model (Durand et al., 2009) showed that the temporal evolution of A yields the hysteretic behavior of the grounding line, implying the need to constrain the range of A. The values of A recent inference of the stress exponent (Ranganathan et al., 2021; Qi and Goldsby, 2021) argues for n of approximately four, especially for highly stressed ice shelf (e.g., Ross Ice Shelf) (Millstein et al., 2022). However, laboratory ice experiments (Durham et al., 1983) proposed an extremely wide range of A from $10^{-35}$ to $10^{-29}$ $Pa^{-4}s^{-1}$ for n = 4. A first-order calculation (i.e., shallow shelf approximation) for the
Antarctic ice velocity estimated A up to $10^{-27}$ $Pa^{-n}s^{-1}$ for n = 4 (Millstein et al., 2022). The higher nonlinearity of ice rheology on the stress at n = 4 (Bons et al., 2018), compared to that at n =3, indicates that the more refined range of A is essential for appropriate viscosity parameterization. Moreover, full Stokes modeling (Durand et al., 2009) demonstrated that the ice mass balance was significantly affected by a change in A in an order of one, even when n = 3. Thus, we attempt to restrict the range of A in the framework of full Stokes modeling to an order of one to two when n = 4.

In this study, we developed a simplified model with a uniform bed slope and an Antarctic bed topography (BedMap2). We performed intensive numerical simulations solving the full Stokes equations (Gagliardini et al., 2013) to analyze the effects of documented fluidity parameters A when n = 4. To verify the modeled ice velocity, we employed the Antarctic ice velocities obtained from Antarctic ice motion assembled from the geodetic data using the multi-satellite interferometric synthetic aperture radar system from 2007 to 2009 provided by NASA's MEaSUREs (Rignot et al., 2011; Mouginot et al., 2012; Rignot et al.,
2017; Mouginot et al., 2017). We found that the simplified and Antarctic models shared the finite range of A that resolved the geodetically observed magnitudes of ice flow velocity. Our study provides insights into accurate parameter selection by constraining the range of rheological parameters in Antarctic ice shelves.



## 2. Methods

### 2.1 Numerical model set up

We built two different models with a simplified slope and the bed topography in Antarctica to verify the effect of fluidity parameters when n = 4, comparing ice flow velocity over a total period of 100 years. In the simplified slope model (Fig. 1a) and Antarctic bed topography model (Figs. 1b-1g), we adopted the simplified bed elevation b(x) from the numerical modeling of ice sheet discharge (Durand et al., 2009; Pattyn et al., 2012) using Eq. (1).

$$\mathrm{b}(x) = 720.0 - 778.5 \times \frac{x}{750 \text{ km}} \;, \tag{1}$$

For the simplified slope model, we constructed initial ice geometries of two-dimensional (2D) models with prograde slope glaciers from an 1800 km (length) × 0.1 km (thickness) ice layer by applying the accumulation of 0.3 m/year for 10,000 years to the ice layer. The initial grounding line was located at x = 1052 km for the simplified slope model (Fig. 1a). In the Antarctic model, the initial grounding lines (yellow triangles in Figs. 1b-1g) were determined by the contact force integrating ice surface, thickness, and bed topography provided by BedMap2. To verify that the modeled grounding lines were consistent with the

Antarctic bed topography, we compared the grounding from BedMap2 (dotted lines).

Friction and hydrostatic pressure boundary conditions were assigned to the lower interface in contact with the bedrock and sea, respectively. Considering the resolution of BedMap2 (~1 km; Fretwell et al., 2013), we employed 0.9 km as the mesh size for the simplified and Antarctic bed topography models, which can minimize interpolation of bed topography during the grid structure construction.

### 2.2 Governing equations

The full Stokes equations were adopted to calculate the ice flow velocity and pressure fields. The Stokes equations comprise the momentum conservation equations (Eq. 2), which assume an incompressible fluid (Eq. 3).

$$-\nabla p + \nabla \cdot \boldsymbol{\tau} = \rho_i \mathbf{g} \tag{2}$$

$$\nabla \cdot \boldsymbol{u} = 0 \text{ where } \boldsymbol{u} = (u, v) \tag{3}$$

$u$ and $v$ denote the velocities in the horizontal ($x$) and vertical ($y$) directions, respectively. $\eta$, $\rho_i$, $\mathbf{g}$, and $P$ indicate the ice viscosity, ice density, gravitational acceleration, and pressure, respectively. The viscosity constitutes the relationship between the deviatoric stress ($\boldsymbol{\tau}$) and the strain rate $(\dot{\boldsymbol{\varepsilon}})$ tensors following:

$$\boldsymbol{\tau} = 2\eta\dot{\boldsymbol{\varepsilon}} \text{ where } \dot{\boldsymbol{\varepsilon}} = \frac{1}{2}(\nabla\boldsymbol{u} + (\nabla\boldsymbol{u})^{\mathrm{T}}) \tag{4}$$

Glen's Law (Eq. 5) determines the viscosity as a function of A value and the power of the effective strain rate $\dot{\varepsilon}_e$,

representing the magnitude of the strain rate tensor.



$$\eta = 2^{-1}A^{-1/n}\dot{\varepsilon}_e^{(1-n)/n} \tag{5}$$

$n$ and A mean the power stress exponent and the ice fluidity parameter, respectively. The ice viscosity was updated iteratively until convergence following Glen's law, in which the viscosity depends nonlinearly on the strain rate calculated from the velocity field.

Along the lower surface of the ice body, nonlinear basal friction, and hydrostatic sea pressure were defined for the numerical nodes grounded to the bedrock and floating above sea level, respectively. The basal shear stress vector $\boldsymbol{\tau_b}$ on the grounded ice was calculated by the non-penetrating Weertman-type friction law (Eqs. 6 and 7).

$$\boldsymbol{\tau}_b = C_w\|\boldsymbol{u}_b\|^{m-1}\boldsymbol{u}_b \tag{6}$$

$$\boldsymbol{u}\cdot\widehat{\boldsymbol{n}} = 0 \tag{7}$$

$C_w$, $\boldsymbol{u}_b$, $m$, and $\widehat{\boldsymbol{n}}$ denote the friction coefficient, sliding velocity, sliding exponent, and outward normal vector, respectively. We assumed the zero magnitude of $\boldsymbol{\tau}_b$ on the lower surface, when the ice was ungrounded from the bedrock or floating in the sea. Sea pressure $p_w$ calculated using Archimedes' law, was applied to the bottom of the ice shelf and the calving front, which were below the sea level (Eq. 8).

$$p_w(y,t) = \begin{cases} \rho_w g(l_w(t) - y), & y < l_w(t) \\ 0, & y \geq l_w(t) \end{cases} \tag{8}$$

$\rho_w$, $l_w$, $y$, and $t$ refer to sea water density, the sea level, and the vertical position of the lower surface, and time, respectively. The detailed values and descriptions of the parameters such as $\rho_w$ and $\rho_i$ are listed in Table 1.

The advection equation with the ice velocity field and flux was solved along the 1D discretized space to model the geometrical evolution of the upper ($j = s$) and lower ($j = b$) surfaces (Eq. 9). The ice body was bounded vertically by two free surfaces: the upper surface between the ice and atmosphere and the lower surface in contact with the bedrock or sea.

$$\frac{\partial y_j}{\partial t} + u\frac{\partial y_j}{\partial x} - v = a_j \tag{9}$$

$a_j$ describes the amount of accumulation/ablation ($a_s$) and basal melting/accretion ($a_b$) at the upper and lower surfaces, respectively. We assigned a uniform distribution of temporally constant accumulation on the upper surface while neglecting basal melting. For a numerically stable time match of the free surface evolution (Eq. 9), we defined the time step as 0.01 years. We examined the effect of a smaller time step size (0.005 years) on ice velocity. The time step size was found to have little

effect on the variability of ice velocity (see Fig. S1 in the Supporting Information), ensuring the reliability of the 0.01 year time step size.

**3 Results**



The initial ice geometry and grounding line position were set in two different models: a simplified model and an Antarctic model with bed topography (BedMap2). We adopted full Stokes ice sheet dynamics software, Elmer/ice, to investigate the effect of values of the fluidity parameter (A) when n = 4 (power-law stress exponent). We selected 11 values of A in logarithmic intervals ranging from $10^{-35}$ to $10^{-29}$ $Pa^{-4}s^{-1}$ compiled from previous studies (e.g., Behn et al., 2021; Millstein et al., 2022).

### 3.1 Simplified slope model

We constructed a 2D simplified bed slope model (e.g., Cheng and Lötstedt, 2020). A comprehensive assessment of ice velocity was conducted over a wide range of A values from $10^{-35}$ to $10^{-29}$ $Pa^{-4}s^{-1}$, when n = 4. We defined a set of logarithmically increasing A values while maintaining a uniform snow accumulation rate of 0.3 m/year (e.g., Hooke, 1976). Fig. 2a shows ice velocity evolution by gravity-driven flow over 100 years for different values of A. In the numerical models, we measured the representative ice velocity at the top of the calving front, which is the rightmost part of the ice shelf. We then compared the modeled ice velocity to the range of a subset of NASA's Making Earth System Data Records for Use in Research Environments (MEaSUREs; Rignot et al., 2011; Mouginot et al., 2012; Rignot et al., 2017; Mouginot et al., 2017) ice velocities from 2007 to 2009 that cover the entire Antarctic (1-1000 m/year; shaded zone in Fig. 2a).The A values between $10^{-32}$ and $251.0 \times 10^{-32}$ $Pa^{-4}s^{-1}$ induced a temporal gradual increase in ice velocity, which ultimately produced ice velocity values consistent with the Antarctic ice velocity. In contrast, a sharp increase in ice velocity, excessive compared to the observation, was derived for the case with A = $10^{-29}$ $Pa^{-4}s^{-1}$. The oscillation in ice velocity in Figure 2a is caused by the interaction with the discrete variations in the position of the grounding line, which is determined by the mesh size. As the grounding line advances, the ice velocity decreases rapidly due to the increase in frictional stress. This dynamics lead to oscillations in the overall ice velocity (see Fig. S2 in the Supporting Information).

We examined the changes in the ice velocity averaged over the 100 years (Fig. 2b). For A = $10^{-29}$ $Pa^{-4}s^{-1}$, the ice velocity was ~2000 m/year, which was higher than the maximum value in Antarctica. This significant difference in the modeled and observed ice velocities highlights the importance of choosing an appropriate A value in the simplified model to ensure that the model accurately captures the observed ice velocity patterns.

### 3.2 Antarctic model with bed topography

We also examined regions containing ice shelves with relatively high velocities within the entire Antarctica. We constructed six models for the West (Ronne, Thwaites, and Ross Ice Shelves) and East (Amery, Shackleton Ice Shelves, and Mertz Glacier Tongue) Antarctica to quantify the effect of different values of A on ice sheet behavior with the realistic bed topography.



We plotted the ice velocity (Figs. 3b-3g) at a model run time of one year for six Antarctic ice shelves (boxes in Fig. 3) with an A value of $16.0 \times 10^{-32}$ Pa$^{-4}$s$^{-1}$, which roughly constrained the Antarctic ice velocity in the simplified model. By self-consistently accounting for the normal force (ice weight) and seawater pressure, the numerical model calculated the initial
grounding lines (yellow triangles), which were similar to the positions of the grounding lines (dotted lines; BedMap2). Our models yielded ice velocities at the calving front ranging from ~0.5 m/year (Fig. 3f) to ~2200 m/year (Fig. 3b). The West Antarctic ice shelf (Figs. 3b-3d) is based on a marine ice sheet system, which reduces the frictional shear stress due to seawater pressure, resulting in a large ice mass flux to the ocean and fast ice flow velocity. However, the models predicted slow ice flow in East Antarctica, where the grounded ice sheet dominates over the marine ice sheet. The range of modeled ice velocities
on the Ross and Amery Ice Shelves are the ~360 m/year and ~570 m/year at the calving front, which are broadly consistent the observed Antarctic ice velocity. However, the velocities calculated from the model with Antarctic bed topography using too small ($10^{-35}$ Pa$^{-4}$s$^{-1}$) or too large ($10^{-29}$ Pa$^{-4}$s$^{-1}$) A values differed significantly from the observations (see Fig. S3 in Supporting Information).

We measured the representative ice velocity at the calving front of the ice shelf model (Fig. 4). By comparing the
time-averaged ice velocities from the model and observations, we attempted to constrain the value of A that adequately explains Antarctic ice velocity. The range of A values derived from the Antarctic ice sheet model for West and East Antarctica was one order of magnitude, showing a significant refinement from the experimentally derived range of six orders. In the Ronne Ice Shelf (Fig. 4a), the values of A between $6.3 \times 10^{-31}$ and $10^{-29}$ Pa$^{-4}$s$^{-1}$ resulted in ice velocities higher than the range of observed averaged ice velocity. Thwaites models (Fig. 4b) with A $> 6.3 \times 10^{-31}$ Pa$^{-4}$s$^{-1}$ showed a faster ice flow compared
to the observation. Similarly, for the Ross Ice Shelf (Fig. 4c), A values of $10^{-35}$ to $0.6 \times 10^{-34}$ Pa$^{-4}$s$^{-1}$ and $6.3 \times 10^{-31}$ to $10^{-29}$ Pa$^{-4}$s$^{-1}$ were lower and higher than observed ice velocities, respectively. For the Amery Ice Shelf (Fig. 4d), the ice flow was faster than the observed for A $> 4.0 \times 10^{-32}$ Pa$^{-4}$s$^{-1}$. For the Shackleton Ice Shelf (Fig. 4e), A values $< 4.0 \times 10^{-32}$ Pa$^{-4}$s$^{-1}$ corresponded to ice velocities lower than the observed. In the Mertz Glacier Tongue (Fig. 4f), A values greater than $6.3 \times 10^{-31}$ Pa$^{-4}$s$^{-1}$ derived ice velocities higher than the observed range.

We compared 100 years time-average of modeled ice velocity on the Ronne, Ross, and Thwaites Ice Shelves of West Antarctica. Fig. 5a shows the range of average ice velocities (gray-shaded zone) from 2007 to 2009 for entire Antarctica obtained from observations. In all models of the West Antarctic ice shelves, the ice velocity generally increased with increasing A (Fig. 5a). For the Ronne (pink bars in Fig. 5a) and Thwaites (yellow bars) Ice Shelves, the modeled ice velocity for eight and nine A values include in the observation range, respectively. For the Ross Ice Shelf (blue bar), ice velocities exceeded the
lower and upper bounds of the observation when A was $< 0.06 \times 10^{-32}$ Pa$^{-4}$s$^{-1}$ and $> 16.0 \times 10^{-32}$ Pa$^{-4}$s$^{-1}$, respectively. For Thwaites Ice Shelf, the relatively small gradient in bed topography (compare Figs. 5c and 5b) possibly induced a low degree of nonlinearity in the viscosity evolution. Nevertheless, for A $> 63.0 \times 10^{-30}$ Pa$^{-4}$s$^{-1}$ the modeled velocity exceeded the upper bound of the observed velocity. Overall, we argue that the A values of $0.06 \times$ to $16.0 \times 10^{-32}$ Pa$^{-4}$s$^{-1}$ satisfied together the criteria for averaged velocities observed in the Ronne, Ross, and Thwaites Ice Shelves. The ice velocities at time



= 100 years for A = $16.0 \times 10^{-32}$ Pa$^{-4}$s$^{-1}$ (gray-shaded area in Fig. 5a) that satisfy all the Ronne (Fig. 5b), Ross (Fig. 5c), and Thwaites (Fig. 5d) Ice Shelves are shown. The shelves derived ice velocities at calving fronts of up to 37 m/year (Ronne), 26 m/year (Ross), and 76 m/year (Thwaites) at time = 100 years.

Ice velocities averaged over 100 years for the Amery and Shackleton Ice Shelves, and the Mertz Glacier Tongue were plotted and compared with the observed ice flow velocities (Fig. 6). For the Amery Ice Shelf models, the time-averaged

velocity generally increased with A values (orange bars in Fig. 6a). Because the floating segment of the Amery Ice Shelf is sufficiently long (647 km of a total length of 1502 km), the ice velocity patterns with different A values are similar with those of the West Antarctic ice shelves. However, the Shackleton Ice Shelf (mint bars) and the Mertz Glacier Tongue (green bars), where the floating segments are relatively short, have lower average velocities than the Amery Ice Shelf. For the Amery Ice Shelf, the model with A > $16.0 \times 10^{-32}$ Pa$^{-4}$s$^{-1}$ yields averaged velocities that exceed the observations. For Shackleton Ice

Shelf models with A values < $4.0 \times 10^{-32}$ Pa$^{-4}$s$^{-1}$, the calculated velocity was outside the range of the observation. Almost all A values except for A > $63.0 \times 10^{-30}$ Pa$^{-4}$s$^{-1}$ matched the observed ice velocities in the Mertz Glacier Tongue.

The ice velocity distribution within the ice sheet at time = 100 years or the Amery (Fig. 6b) and Shackleton (Fig. 6c) Ice Shelves and the Mertz Glacier Tongue (Fig. 6d) are shown with A values of $16.0 \times 10^{-32}$ Pa$^{-4}$s$^{-1}$ that satisfied all of three ice sheets (dotted line in Fig. 6a). The ice velocities measured at the calving front of the ice shelf at time = 100 years were 415,

1, and 16 m/year for the Amery, Shackleton Ice Shelves, and Mertz Glacier Tongue, respectively. We argue that the A values satisfying the simplified and Antarctic bed topography are between $4.0 \times$ and $16.0 \times 10^{-32}$ Pa$^{-4}$s$^{-1}$, which is narrower (order of one) than that suggested by previous studies (order of six).

## 4 Discussion

In this study, we quantified the effect of Glen's law parameter A (fluidity) over a wide range on ice sheet dynamics by

comparing ice velocity, from a simplified slope and the Antarctic bed topography models. Recent ice sheet dynamics models have argued that Glen's law with n (power-law stress exponent) = 4, instead of n = 3, better explains ice flow in Antarctica (e.g., Behn et al., 2021; Ranganathan et al., 2021). The ice viscosity at n = 4 depends more nonlinearly on the strain rate and stress than that at n = 3. For example, the ice viscosities at n = 4 and n = 3 were reduced by factors of ~200 and ~100, respectively (see Eq. 5), over the range of strain rates (1 to 100 s$^{-1}$). The higher dependence of ice rheology on the strain rate

at n = 4 indicates that constraining the values of A is essential for robust ice sheet dynamics modeling.

Previous estimates of Glen's Law parameters have mostly employed models with only the floating ice shelf (Kirchner et al., 2011; Millstein et al., 2022). The ice shelf-only model yields a simple stress regime occurred in which the strain rate is approximately constant with ice depth due to the negligible basal drag at the ice sheet base (Ranganathan et al., 2021). However, we considered the basal drag along the grounded parts, so the strain rate varied strongly with depth. The difference between

the stress patterns of the upper and lower ice surfaces was significant, resulting in stress transfer to the adjacent region. In



addition, the basal drag leads to the complexity near the transition zone between the ice shelf and sheet, where stress changes abruptly, affecting viscosity under the nonlinear Glen's law (Davis et al., 2023; Holland et al., 2023).

Because we employed a continental-scale model from the ice divide to the calving front, our model better constrains the A values that affect glacier dynamics when n = 4, based simultaneously on ice sheet drag and hydrostatic pressure on the
calving front. Our model with a large spatial extent and basal topography also constrains the Antarctic ice velocity in the n = 4 case for a limited range of A values from $4.0 \times$ to $16.0 \times 10^{-32}$ $Pa^{-4}s^{-1}$. The n = 3 model of Antarctic ice sheet dynamics by Durand et al. (2009) set A values that differ by less than a factor of 100, causing grounding line migration of more than 10 km. Thus, for Glen's law with n = 4, which is more nonlinear than for n = 3, the narrower range of A values (within an order of one) needs to be constrained for precise calculation of Antarctic ice velocity and grounding line migration.

Accurate numerical simulations of ice sheet dynamics require proper application of the interaction of ice viscosity with other dynamics, such as friction and hydrostatic water pressure (Golledge et al., 2019; Rasmussen and Thomsen., 2021). We adopted the Weertman friction law to define the relationship between ice sliding velocity and basal shear stress (Gagliardini et al., 2007; Barnes and Gudmundsson, 2022). The friction laws (e.g., Schoof and Budd laws) include normal stress, subglacial flow pressure, and cavitation (e.g., Gagliardini et al., 2007), and ice thickness control the normal stress and subsequent effective
basal shear stress. The nonlinear dependence of the ice basal velocity on the normal stress, and thus the selection of an appropriate friction law for different regions, is essential.

Varying the sliding exponent (m in Eq. 6), which controls the nonlinearity of the basal shear stress on the sliding velocity in the Weertman friction law, strongly influenced the ice velocity. This implies that the dependence of shear stress on sliding velocity is predominant over normal stress in continental-scale ice sheet dynamics. For the Pine Island Glacier, the value of m
was estimated to be >20, which is much larger than the conventional value of m (i.e., ~3; Brondex et al., 2017) to explain the observed surface velocities. Models with a larger value of m show a faster response of the ice sheet to changes in boundary forcing (e.g., ice thickness change by accumulation and basal melting) (Brondex et al., 2019). For ice sheet models with a stress exponent n = 4, the effect of the sliding exponent on the ice velocity was less quantified than that for the models with n = 3. For the n = 4 models, viscosity decreased more sensitively with increasing strain rate than for the n = 3 models. As ice
velocity increased, frictional shear stress increased, which regulated ice velocity. Thus, the ice sheet model with n = 4 is necessary to reflect the friction law with an appropriate sliding exponent. For the Antarctic geometry, we changed the value of m = 3, 5, 7, and 9 (see Fig. S4) (Barnes and Gudmundsson, 2022) while keeping the ice rheological parameters fixed (i.e., n = 4). The variation of ice velocity with the value of the sliding exponent (m) showed a relatively small difference (less than 10%) compared to the fluidity parameter (A) of Glen's law.

**5 Conclusions**

Understanding the complex dynamics of ice sheets is critical to interpreting the past and predicting the future associated with ice mass loss and subsequent sea level rise. The stress exponent (n) related to the deviatoric stress in Glen's law implies



that ice viscosity becomes more nonlinear as the value of n increases from three to four. Increasing nonlinearity magnifies the effect of another parameter of Glen's law, the fluidity parameter (A), which has important implications for modeling ice sheet dynamics. The value of A is determined by the interaction of various environmental factors within the glacier, such as temperature, water content, and pressure. We focused on evaluating the effects of different values of A on ice flow velocity, with n = 4. We then attempted to narrow the range of A to explain the Antarctic ice velocity from the extremely wide range derived from the laboratory experiments (an order of six). Based on both a simplified slope and Antarctic bed topography models, we narrowed the range of A (an order of one) to constrain Antarctic ice velocity. We identified a critical range of A values, from $4.0 \times$ to $16.0 \times 10^{-32}$ $\mathrm{Pa}^{-4}\mathrm{s}^{-1}$, accurately representing ice velocity relative to the observed ice velocities averaged from 2007 to 2009 in Antarctica (MeaSUREs). Modeling ice sheet behavior as a function of fluidity parameters improves the ability to predict ice sheet response to changing climate conditions, ultimately leading to accurate assessments of future sea level rise.

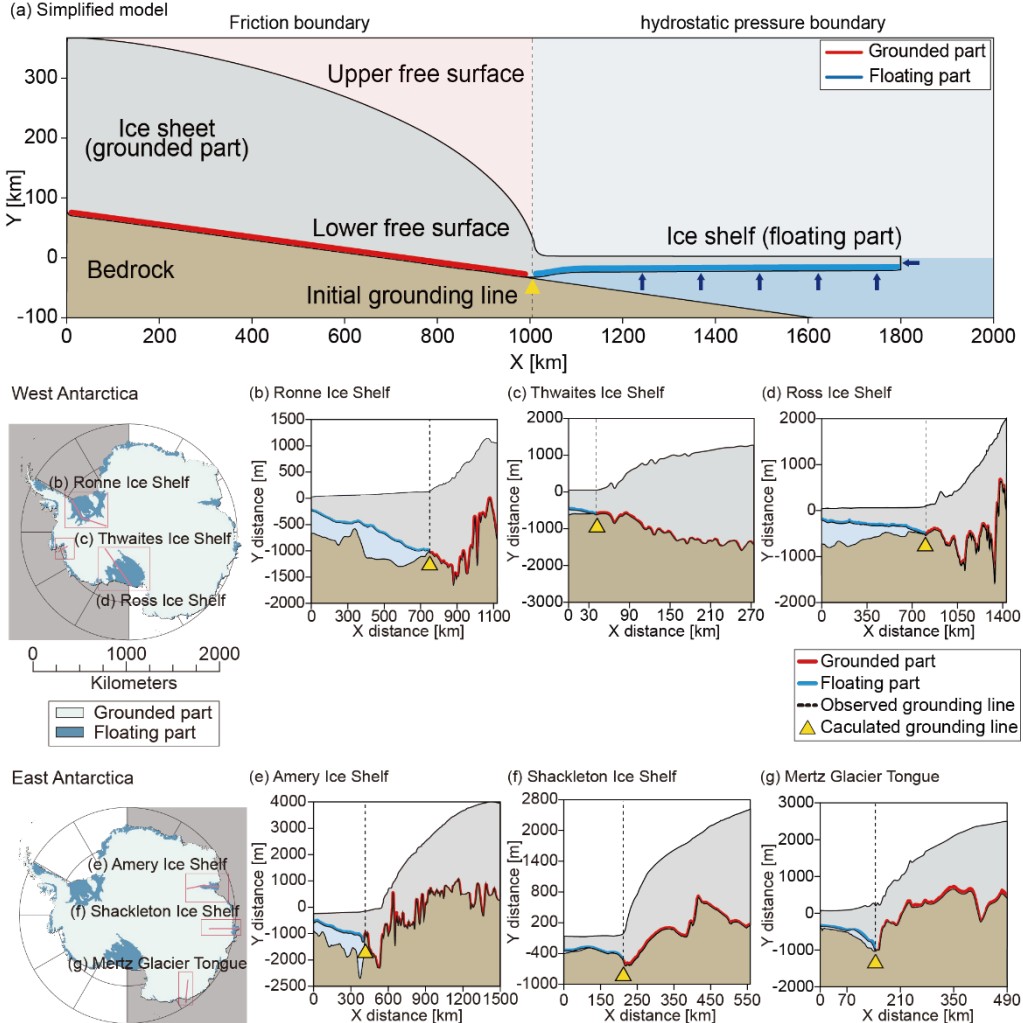


**Figure 1.** (a) The simplified slope model. The initial ice sheet geometry was created by the accumulation of 0.3 m/year for 10,000 years on a long and thin rectangular ice sheet (1800 km × 10 m). The initial grounding line, self-consistently determined by contact force, divides the ice sheet into two segments: the grounded (red line) and the floating (blue line). Frictional and hydrostatic water pressure boundary conditions are applied to the grounded and floating segments, respectively. (b-g) Antarctic

models constructed from bed topography, ice surface, and ice thickness. A total of six representative ice shelves in the West (Ronne, Thwaites, and Ross Ice Shelves) and East (Amery Ice Shelf, Shackleton Ice Shelf, and Mertz Glacier Tongue). The (b) Ronne, (c) Thwaites, and (d) Ross Ice Shelves have total lengths of 1,124, 275, and 1,428 km, respectively. The East Antarctic models target the (e) Amery, (f) Shackleton Ice shelves, and (g) Mertz Glacial Tongue with model lengths of 1,502, 560, and 489 km, respectively. Yellow triangle and black dotted lines refer to the grounding lines from our modeling (solving

the full Stokes equations) and BedMap2, respectively.





**Figure 2.** For different values of A, comparison of (a) time-dependent and (b) time-averaged ice velocities over 100 years at the calving front derived from the simplified model with the range of time-averaged ice velocities in Antarctica (shaded area). When A ranges from $1.0 \times$ to $251.0 \times 10^{-32}\,\mathrm{Pa}^{-4}\mathrm{s}^{-1}$, the calculated time-dependent and time-averaged ice velocities are within the range of observations in Antarctica. (c-e) Ice velocity at time =1 year with $63.0 \times 10^{-32}\,\mathrm{Pa}^{-4}\mathrm{s}^{-1}$, $251.0 \times 10^{-32}\,\mathrm{Pa}^{-4}\mathrm{s}^{-1}$ (within the gray shaded zone in a and b) and $10^{-29}\,\mathrm{Pa}^{-4}\mathrm{s}^{-1}$. (f-h) Comparison of ice velocities at time = 100 years.





**Figure 3.** In (a) six regions (boxes) containing the selected ice shelves, (b-g) the distributions of ice velocity at the model elapsed time = 1 year when A = $1.5 \times 10^{-31}$ Pa$^{-4}$s$^{-1}$ and n = 4. (b) Ronne, (c)Thwaites, and (d) Ross Ice Shelves of West Antarctica, whereas (e) Amery, (f) Shackleton Ice Shelves, and (g) Mertz Glacier Tongue of East Antarctica. The yellow triangles indicate the self-consistent grounding line calculated by solving the Stokes equations. The complex interplay of factors such as bed topography, ice thickness, and seawater pressure contribute to the calculation of ice velocity,







**Figure 4.** Time-dependent ice velocities derived from the models of (a) Ronne, (b) Thwaite, and (c) Ross Ice Shelves in West Antarctica and d) Amery, (e) Shackleton Ice Shelves, and (f) Mertz Glacier Tongue in East Antarctica, plotted with different A values (colors of lines). The shaded area represents the range of observed ice velocities averaged from 2007 to 2009 for the Antarctic ice shelves. A finite ranges of A values were identified for all models that produced calculated velocities matched with the observed data.




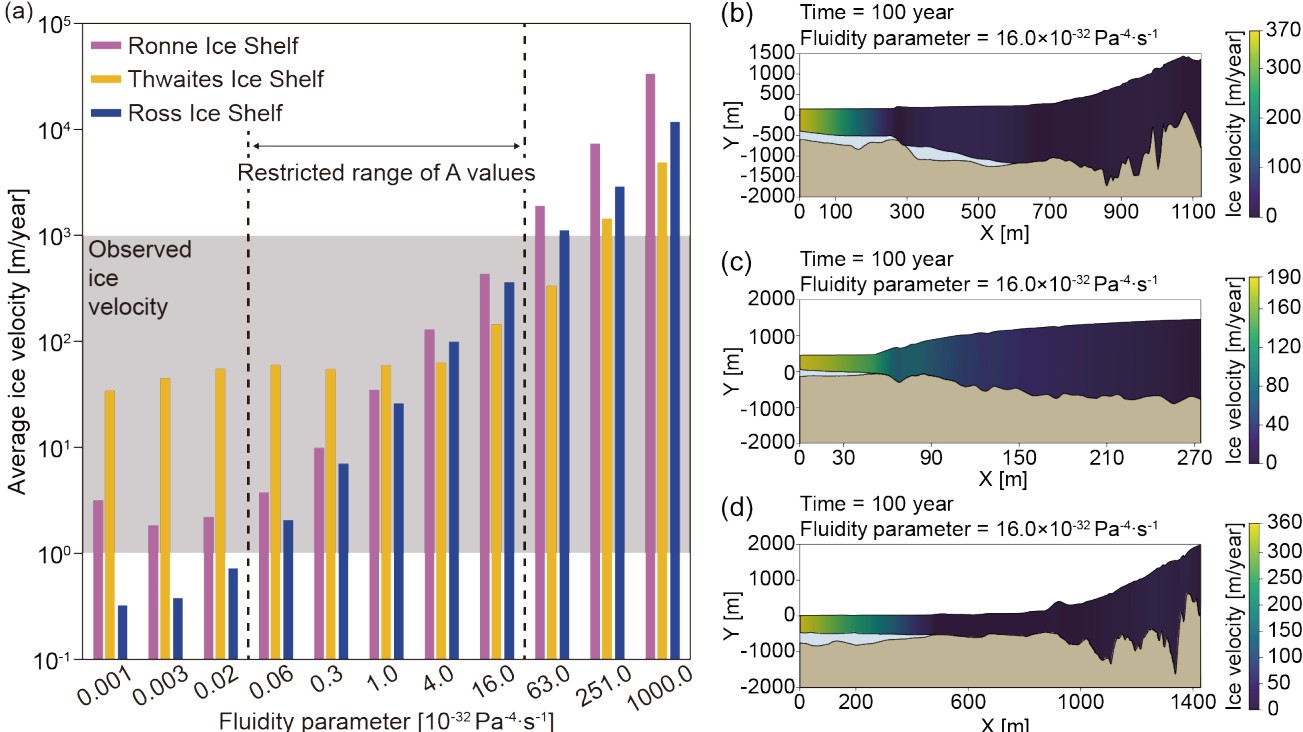

**Figure 5.** (a) The 100 year time-averaged ice velocities of the Ronne (purple bar), Thwaites (yellow bar), and Ross (blue bar) Ice Shelves calculated for eleven different A values ranging from $10^{-35}$ to $10^{-29}$ $Pa^{-4}s^{-1}$. The gray-shaded area represents the range of observed Antarctic ice velocities. The dotted line marks the range of A values at which the modeled ice velocities lie within the gray-shaded area. The distributions of ice velocities at time = 100 years of (b) Ronne, (c) Thwaites, and (d) Ross Ice Shelves are plotted over the initial configurations. A value of $16.0 \times 10^{-32}$ $Pa^{-4}s^{-1}$ was chosen, which is included simultaneously in the gray area and the dotted line.




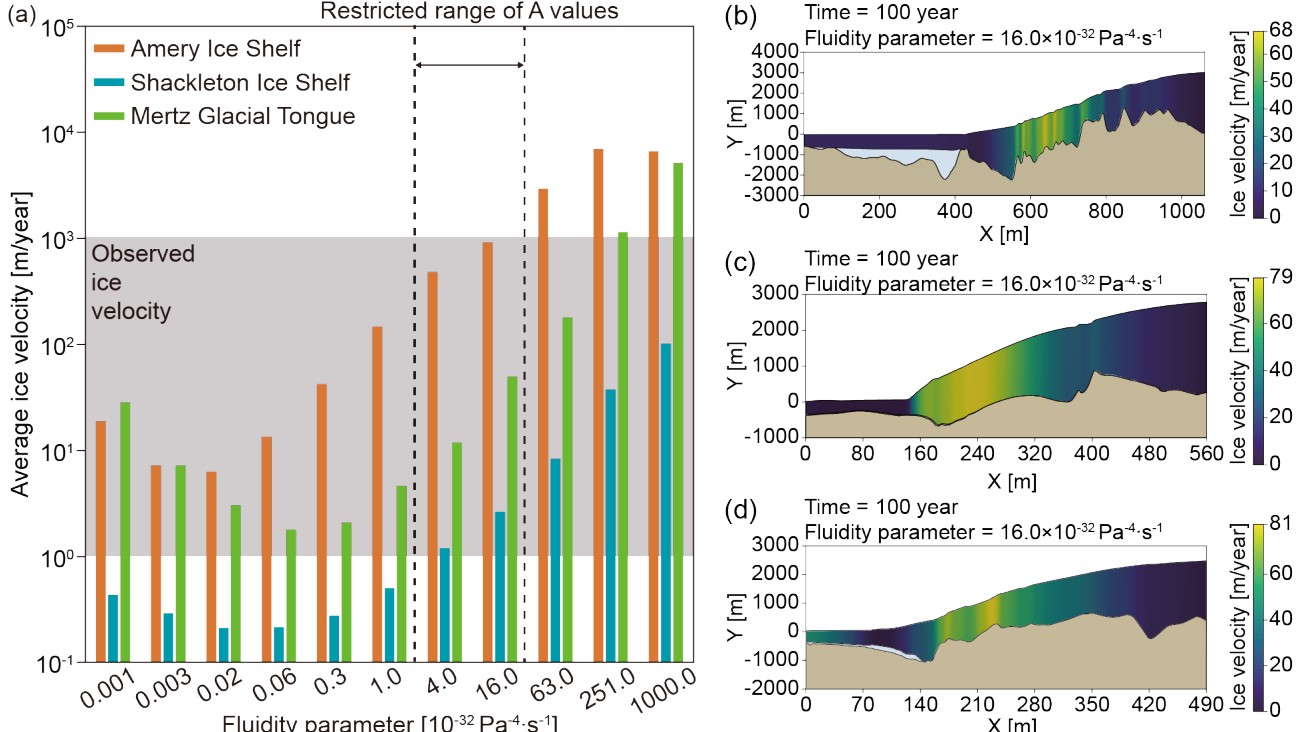

**Figure 6.** (a) The time-averaged ice velocities, measured at the calving front, for the Amery (orange bar) and Shackleton (mint bar) Ice Shelves and Mertz Glacier Tongues (green bar), derived from modeling with eleven different A values ranging $10^{-35}$ to $10^{-29}$ Pa$^{-4}$s$^{-1}$. The gray-shaded zone denotes the range of average Antarctic ice velocities. The dotted lines indicate the range of A values for which the modeled ice velocities coincide with the gray zone. (b) The Amery and (c) the Shackleton Ice Shelves, and (d) Mertz Glacier Tongue were modeled with an A value of $16.0 \times 10^{-32}$ Pa$^{-4}$s$^{-1}$.



**Table 1.** Values and Descriptions of parameters

| Description | Symbols | Values [unit] and equations |
|---|---|---|
| Ice density | $\rho_i$ | 900 [kg/m$^3$] |
| Sea water density | $\rho_w$ | 1000 [kg/m$^3$] |
| Sea level | $l_w$ | 0 [m] |
| Accumulation rate | $a_s$ | 0.3 [m/year] |
| Fluidity parameter | $A$ | $10^{-35}$ to $10^{-29}$ [Pa$^{-4} \cdot$ s$^{-1}$] |
| Friction coefficient | $C_w$ | $7.624 \times 10^6$ [Pa $\cdot$ m$^{-1/3} \cdot$ s$^{1/3}$] |
| Sliding exponent | $m$ | 3, 5, 7, and 9 [-] |
| Strain rate | $\dot{\varepsilon}_{xx}, \dot{\varepsilon}_{yy}$ | [1/s] |
| In-plane strain rate | $\dot{\varepsilon}_{zz}$ | $-\left(\dot{\varepsilon}_{xx} + \dot{\varepsilon}_{yy}\right)$ [1/s] |
| Effective strain rate | $\dot{\varepsilon}_e$ | $\left(0.5\left(\dot{\varepsilon}_{xx}^2 + 2\dot{\varepsilon}_{xy}^2 + \dot{\varepsilon}_{yy}^2 + \dot{\varepsilon}_{zz}^2\right)\right)^{0.5}$ [1/s] |

*Author contributions*

SJL and BDS organized the original draft preparation, conceptualization, validation, and Formal analysis. BDS performed Supervision and Funding acquisition. YJB provided review & editing.

*Data Availability Statement*

We use Elmer/Ice, the glaciological extension of the open-source finite-element code Elmer (www.csc.fi/elmer) (Gagliardini et al., 2013). We adjusted bed topography, ice surface, ice thickness, and grounding line position in BedMap2 (Fretwell et al., 2013; https://www.bas.ac.uk/project/bedmap-2). Figures were generated using QGIS (QGIS Development Team, 2021). All relevant data for the refinement of fluidity parameters are available on Zenodo: https://doi.org/10.5281/zenodo.10148353.

*Competing interests*

 The authors declare that the research was conducted in the absence of any commercial or financial relationships that could be construed as a potential conflict of interest.





*Acknowledgements*

This work was supported by the National Research Foundation of Korea (NRF) grant funded by the Korea government (MSIT) (NRF-2022R1A4A3027001) and Korea Institute of Marine Science & Technology Promotion (KIMST) funded by the Ministry of Oceans and Fisheries (RS-2023-00256330), which was awarded to B-D S. S-J L. and Y-J B. was partly supported by the Ministry of the Interior and Security as a human resources development project in the field of disaster management.

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
