# Peer review of "Refinement of the fluidity parameter range with a stress exponent of four in Glen's law: insights from Antarctic bed topography model"

_EGUsphere, 2023_

## Referee Comment (RC1)

**Review of Refinement of the fluidity parameter range with a stress exponent of four in Glen's law: insights from Antarctic bed topography model**

**General Comments**

Based on recent work (Millstein et al., 2022; Ranganathan et al., 2021) there is increasing interest in using n=4 rather than n=3 in ice sheet modelling. This paper uses 2D full-Stokes simulations with n=4 to provide order-of-magnitude constraints on the value of the fluidity parameter when n=4. They reduce the range of A from 6 orders of magnitude to 1. Generally, I think the paper provides a useful contribution to the community and should be published after the inclusion of a few more clarifying figures and paragraphs.

Originality: The paper performs simple 2D simulations of regions of ice sheets. The novelty comes from using the value of n = 4 and exploring the range of A values in the rheology which give reasonable results. The main contribution is new constraints on the value of A to use when using n=4. This will be useful to ice sheet modellers interested in using this value of n, of which there is increasing interest in.

Scientific Quality: The authors use 2D full-stokes simulations to constrain appropriate value of A which give predictions which match observed velocities. The use of full-Stokes is good, and I am also pleased the authors demonstrate that the results are robust to a range of sliding exponents.

One concern I have about the final range of A values given is that it may over-constrain A. The final range of A values stated, 4 to $16 \times 10^{-32}$ $Pa^{-4}s^{-1}$ , is found by simply combining the A values which give reasonable results for each case studied. One could imagine that if more ice shelves were included in this study, this method would give no A value which gave reasonable results for all. This is not a criticism of the work, which I think is valid, but just the calculation procedure used at the end.

The Shackelton Ice shelf in particular is an outlier in terms of the appropriate range of A values. It would be useful to explore in the discussion what the suggested range of A values would be without this case and have some statistical analysis and a final summarizing figure showing the suggested A values for each case examined.

There is also likely sufficient unaccounted for physics implicit in A, such as temperature variation, fabric induced anisotropy, grain size etc. which would give an order of magnitude variation at least in the true value. This should also be mentioned in the discussion.

Significance: The paper will be useful to ice sheet modellers in general, and hopefully encourage more studies using n=4.

Presentation quality: The paper is generally well written and concise. The title provides a clear summary of the work, and the abstract communicates the main results, though I think it should be noted in the abstract that this is a 2D study.

**Specific Comments**

- I would like to see at least one 3D simulation alongside the 2D cases for the idealised configuration explored here, to ensure that the same range of A values are found in 2D and the

3D cases. The limitation of 2 dimensions and how things may change in 3D should also be included in the discussion.

- There should be a demonstration of the mesh independence of the results, especially considering the saw-tooth oscillations in Fig 2., for both the idealised and realistic cases. This can just be included as a supplementary figure.
- The definition in Eq. 6 and your values of m = 3,5,7,9 later are inconsistent: with these m values Eq. 6 should read 1/m not m.

**Technical Comments**

Line 30-31: By incorporating geodetic creep data… I'm not sure what is meant precisely by this phrase. Furthermore, while the Cuffy and Behn do support n=3, it would be relevant to say here that n=3 has been used since Glen's classic experiments (Glen, 1952).

Throughout the manuscript mathematical variables in the text should be written in math mode, i.e. $A$ and $n = 4$ rather than A and n=4.

Line 85: should be lowercase p for pressure

Around Eq. 4 and 5 $\varepsilon_e$ should be defined or the table should be referenced

The range of Fluidity parameters in Fig 2 should be expressed in exponent notation (i.e. 10^-29 rather than 0.001 x 10^-32)

Line 150 of the text references dotted lines showing the positing of BedMap2 grounding lines in Fig 3, yet I cannot see them in this figure.

---

## Author Comment (AC1)

**Response Letter to Reviewer #2**

Blue letters: revised and/or added text in the revised manuscript

 with strikethrough: deleted text in the revised manuscript

Text in *italic*: comments from Reviewer #2

*This study seeks to constrain the values of the fluidity parameter A when using a stress exponent value of n=4, rather than the canonical n=3 value. The authors use a full Stokes model in two configurations – a simplified, linear bed slope model and a model of Antarctic bed topography – with varying values of A and n=4 and compare the resulting velocities to those observed in Antarctica. The authors ultimately constrain the value of A to a small range of 4 x 10^-32 to 16 x 10^-32 Pa^-4 s^-1. The goal of this work is an important one, as it makes the use of n=4 accessible in ice sheet models by providing a calibrated value of A to use alongside n=4. Further, constraints on the fluidity parameter inherently provide constraints on many physical properties of ice sheets, stated by the authors, that can affect future flow. I have questions about the methodology that would benefit from further analysis and explanation in the paper for the readers to derive insight from the results in this study. I would recommend that those questions be explored prior to publication.*

**Authors' reply:**

We thank Reviewer #2 for the valuable feedback. The purpose of our research is to constrain the generalised range of the fluidity parameter (A value) when changing the value of the stress exponent (n) from the conventional 3 to 4. We believe that our study can provide the potential for more accurate simulation of complex ice flow and glacier topography when using n = 4 in ice sheet models. We have improved our manuscript based on the reviewer's suggestions. We have included our response and related documentation, such as figures and references, in this response letter.

*Methodology:    The method of prescribing a constant fluidity value across the domain (of both the simplified model and the Antarctic models), and then varying this value to match observations seems to have some limitations. Firstly, the constant A field inherently limits the takeaways of this study, as fluidity is likely to vary spatially (and temporally) due to a number of ice properties, such as temperature, ice damage, ice crystalline fabric, etc. Deviations between the 1odelled velocities and the observed velocities could be due to other factors besides the average A field, such as small regions of elevated A values (due to, perhaps, damage and fracturing), or variations in temperature at depth. Secondly, the method is restricted by the specific values that the authors choose to evaluate. It makes me wonder why the authors didn't apply a formal inversion method (such as Larour and others, 2005, among many others that have used such a technique) with n=4. Such a method would be able to capture spatial variations in A, at least in a two-dimensional sense, and would not be restricted by the values chosen by the authors. At the very least, the authors should add to the discussion section a description of these simplifications and the implications for these results.*

**Authors' reply:**

We sincerely appreciate the reviewer's insightful feedback and the opportunity to discuss the methodological approach of our manuscript. The reviewer's comments on the use of inversion methods to estimate rheological parameters in modelling ice sheet dynamics have prompted a thoughtful revision of our research focus and methodology. Inversion methods have been widely used to estimate the distribution (in terms of initial conditions or data assimilation) of important parameters (e.g., the A value for Glen's law and C value for Weertman friction law) (Choi et al., 2023; McArthur et al., 2023) that are difficult to measure directly. We fully agree that the inversion method advances the understanding of ice sheets, glaciers, and their interactions with climate change.

[Figure]

**Figure 10.** Evolution of basal stress between 1985 and 2018. First column: basal stress from the static friction coefficient and viscosity parameter experiment (TI_CTR1). **(a1)** Basal stress in 1985 and **(b1, c1, d1)** changes in basal stress, compared to 1985, in indicated years. Second column: **(a2)** basal stress in 1985 and **(b2, c2, d2)** its change for the transient friction coefficient and static viscosity parameter experiment (TI_CTR2). Third column: **(a3)** basal stress in 1985 and **(b3, c3, d3)** its change for the static friction coefficient and transient viscosity parameter experiment (TI_CTR3). Last column: **(a4)** basal stress in 1985 and **(b4, c4, d4)** its change for the transient friction coefficient and viscosity parameter experiment (TI_CTR4).

Impact of time-dependent data assimilation on ice flow model initialization and projections: a case study of Kjer Glacier, Greenland (Choi et al., 2023)

Choi et al. (2023) used the inversion method to estimate the basal friction coefficient (C value) and fluidity (A value) from observations (e.g., ice velocity, ice front position, bed topography, and ice thickness). Choi et al. (2023) introduced two different inversion methods: the "snapshot inversion method" and the "time-dependent inversion method". Each method derived different C and A values. Forward models, using C and A values derived by the snapshot inversion method, accurately reproduced the state at a given time, but failed to predict temporal changes such as acceleration of the ice flow. The model consistently underestimated velocities compared to actual observations. Conversely, the time-dependent inversion method, which incorporates changes over time using historical data, allowed relatively accurate predictions of ice velocity changes, especially ice acceleration. This discrepancy in the simulation results due to the different inversion approaches suggests that A values may vary depending on the inversion method. Therefore, we believe that it would be beneficial to objectively test a wide range of constant A values for our study, which aims to approximate the general range of A values.

Wolovick et al. (2023) also employed the inversion method to investigate basal friction beneath the Antarctic Filchner-Ronne Ice Shelf. In particular, the study by Wolovick et al. (2023) focused on determining the optimised level of normalisation through L-curve (trade-off curve) analysis. The L-curve is used to optimise the regularisation parameter (λ) for the inversion (Aster et al. 2012).

[Figure]

Regularization and L-curves in ice sheet inverse models: a case study in the Filchner–Ronne catchment (Wolovick et al., 2023)

In the left figure above, Wolovick et al. (2023) compared the L-curves for different values (m = 1, 3, and 5) of the sliding exponent (m) when the base friction coefficients of the ice sheet were inverted. The focus was on comparing linear (m = 1; black lines) and nonlinear (m = 3 and 5; green and magenta lines) Weertman friction laws using the L-curve. Wolovick et al. (2023) showed that the nonlinear Weertman friction law produced a steeper limb in the L-curve, resulting in a sharper corner (compare black and magenta lines). This suggests that a small change in the regularisation parameter (i.e., λ) in the nonlinear Weertman law can lead to overfitting and underfitting, which result in an overly constrained C value for the noise-like signal and an overly simple C value distribution, respectively. This suggests that the C value obtained from the inversion method is sensitive to the nonlinearity of the chosen law and the regularisation parameter.

Despite the proven benefits of inversion methods, our study has attempted to purchase an implication of using a stress exponent (n) of 4 with the A value across different modelling scenarios (e.g., idealised slope and Antarctic bed topography models). Our approach of investigating the constant A value over the entire ice shelf, specifically under the n = 4 stress exponent condition, was motivated by the need to contribute to a fundamental understanding of the behaviour of this parameter in ice dynamics models. For n = 3, the A value already has a narrow range in many models through extensive investigation. However, the A value for n = 4 has been studied less intensively.

We present a previous study of ice sheet dynamics modelling using an idealised geometry with n = 3 and constant A values. The idealised model developed by Pattyn et al. (2012) is based on the full Stokes ice sheet flow (see figures below), with various parameters such as A value, C value, and accumulation rate. In addition, Pattyn et al. (2012) also introduced contact dynamics to accurately model basal friction, stress, and viscosity. We cautiously believe that the inversion method to be site-specific. Rheological and frictional parameters (e.g. the A and C values) are inverted based on observed data (e.g. bed topography, ice flow velocity, heat flux) from specific ice shelves. These parameters are then used in the modelling process, which ensures consistency between observed and modelled data. The constant A method may provide a comprehensive and general approach that supports the flexibility to apply the model to different regions and conditions of the ice sheet. In addition, the constant A value allows ice physics to be modelled effectively even with limited observational data. While the inversion method relies on observed data, the use of a constant A allows the model to be applied under a wider range of conditions.

[Figure]

Table 3. Values of $A$ used in EXP 1.

| Step no. | $A$ (s$^{-1}$ Pa$^{-3}$) |
|---|---|
| 1 | $4.6416 \times 10^{-24}$ |
| 2 | $2.1544 \times 10^{-24}$ |
| 3 | $10^{-24}$ |
| 4 | $4.6416 \times 10^{-25}$ |
| 5 | $2.1544 \times 10^{-25}$ |
| 6 | $10^{-25}$ |
| 7 | $4.6416 \times 10^{-26}$ |
| 8 | $2.1544 \times 10^{-26}$ |
| 9 | $10^{-26}$ |

Fig. 1. Results of EXP 1 and 2 performed with a participating FGH model (Docquier et al., 2011) with $\Delta x = 12.5$ km. The different panels show the steady state ice sheet profiles (EXP 1 = advance, EXP 2 = retreat – note that the profiles of both advance and retreat overlap) with the red profile the final profile at the end of EXP 2, the position of the grounding line in time, steady state grounding line positions as a function of the forcing viscosity ($A^{-1}$), and the forcing of $A^{-1}$ over time.

Example of constant A value for entire numerical domain (Results of prograde slope model. Pattyn et al., 2012)

We recognise the importance and effectiveness of inversion methods in improving the fidelity of ice dynamics models. However, by focusing on the constant A value approach for an n = 4 stress exponent, we aim to address a gap in the ice sheet rheology and its implications for ice velocity modelling. Furthermore, our research provides a necessary stepping stone for future studies that may incorporate inversion methods to investigate the spatial variability of A values with n = 4. Our results can be used to set more realistic initial A values when performing inversions for the spatial distribution of A values. We also believe that the narrowed range of A values in our study may be useful as a prefactor when establishing an exponential relationship with temperature, i.e., the fluidity parameter $\propto A_0 \exp\left(-\frac{E_a}{RT}\right)$. Thus, we believe that our study contributes valuable perspectives to the field of ice sheet dynamics and encourages further exploration of its complex rheological properties. We have discussed the advantages and limitations of using constant A values in the revised manuscript as follows.

**In the Discussion section,**

The use of a constant value of the fluidity parameter ($A$) throughout the model domain of the idealised slope and Antarctic bed topography was a methodological choice to simplify the complex interactions inherent in ice sheet dynamics for isolating the effect of the $A$ value. This approach provided a controlled modelling situation to quantify the response of ice flow to different values of $A$. However, our method has the limitation of simplifying local variations in ice properties such as temperature, ice damage and ice crystal fabric that are relevant to ice sheet dynamics.

$\vdots$
$\vdots$

In ice sheet dynamics modelling, the inversion method has been widely used to derive spatially and temporally variable rheological and frictional parameters, as highlighted in previous studies (Choi et al., 2023; Wolovick et al., 2023). In addition to improving the ability of models to fit observed ice velocities, such methods ultimately enhance the predictive capability of future ice behaviour. However, we focused on the use of a constant value of $A$ with a stress exponent ($n$) of 4. The constant value of A provides a baseline value for the sensitivity of ice flow to rheological changes. The narrowed range of $A$ values in our study was achieved by minimising the error between the numerical model and observed Antarctic ice velocity, allowing the range of $A$ values to be used as initial values of temperature-dependent ice viscosity. We expect that this approach will provide insights for the development of refined model parameters.

*The simplified slope model seems to be the most limited in comparisons to observations. How do we know that velocities that differ from observed Antarctic velocities are not due to the simplified nature of the bed geometry?*

*I also wonder if the results from the simplified slope model add to the results that are produced from the Antarctic model, for which the comparison to observations can provide more insight.*

**Authors' reply:**

When studying complex systems such as the Antarctic ice sheet, the ice sheet dynamic models that include all the interactions between all parameters can obscure the important physics. By using a simplified model, we can improve our understanding of the physical processes (e.g. calving and melting/accumulation) separately. Previous studies (Pattyn et al., 2012) have shown that models with simplified geometry have an advantage in quantifying the effects of a particular physics. For example, the Marine Ice Sheet Model Intercomparison Project (MISMIP) is an effort to assess the instability of marine ice sheets (Pattyn et al., 2012). MISMIP has been widely used to investigate the essential elements that determine ice stability (e.g., the dependence of water pressure on sea level, the friction law, viscosity) (Durand et al., 2009; Gladstone et al., 2018). These studies are also based on simple two-dimensional models with prograde and retrograde slopes.

Pattyn et al. (2012) demonstrated that temporal changes in the A value (note that the A value is spatially uniform; see table below) can induce a complex hysteresis motion in the grounding line (see figure below), even in models with a simple slope (prograde and retrograde). This shows that the grounding line can be sensitive to variations in the A value. In models with complex bed topography and inverted A and C values, it is difficult to distinguish whether the occurrence of hysteresis motion is due to the complexity of the bed topography or to the temporal changes in the A value.

[Figure]

Results of the prograde and retrograde slope model. Spatially uniform constant A value (Pattyn et al., 2012)

An idealised slope model was also used in Favier et al. (2012) to analyse the effect of specific processes, such as pinning point, on grounding line dynamics. Simulations were performed with sets of bump   topography from the bed topography (see left figure below). The parameters applied in the calculations were spatially uniform constant values for both friction (C value) and ice viscosity (A value). The present of pinning point causes upstream growth and advance as the thickness of the ice at the grounded part increases. This suggests that the contact at the pinning point acts as a marine ice sheet instability. This approach of using simple models to understand complex dynamics is important in comparative analyses of glacier dynamics when trying to isolate the effects of specific physics. This simplification provides a basis for researchers to clearly understand the different physical elements and their interactions. Especially in systems with many variables and complex processes, such as ice sheet dynamics, idealised slope models can improve our understanding of individual elements.

[Figure]

A three-dimensional full Stokes model of the grounding line dynamics: effect of a pinning point beneath the ice shelf (Favier et al., 2012)

The main objective of our study is to propose an appropriate range of A values for n = 4. Therefore, it was crucial that we use an idealised slope model to test whether the A value consistently regulates ice velocity, regardless of the complex bed topography. By testing the effects of A value changes in an idealised slope model and isolating the A value effect, we could improve the reliability of the Antarctic bed topography model. In our study, we used one of the MISMIP models for Antarctic ice sheet stability studies, the prograde slope ($y$ [km] = 720.0 [km] − 778.6 [km] × $\left(\frac{x\,[\text{km}]}{750.0\,[\text{km}]}\right)$). Consequently, the constrained range of A values from the simple slope models was determined to be $1.0 \times$ to $251.0 \times 10^{-32}\text{Pa}^{-4}\text{s}^{-1}$ (see red shaded zone in the figure below). This range fully includes the constrained range of A values from $4.0 \times$ to $16.0 \times 10^{-32}\text{Pa}^{-4}\text{s}^{-1}$ based on Antarctic bed topography (see dotted lines in the figure below).

[Figure]

Comparison of the range of A values derived from the Antarctic model and the simplified model.

Thus, we found that the idealised slope model derived a wider range reflecting only the influence of the A value. The additional influence of the detailed bed topography then further narrowed the A value range. In the original manuscript, we considered, without a deep understanding, that the adequacy of the A value was ensured by constraining both the simple slope model and the bed topography model. However, thanks to the careful advice of Reviewer #2, we can discuss the value of constraining the Antarctic A values in a simple slope context (isolating the effect of the A value from the bed topography). We have included this discussion in the revised manuscript as follows.

In the Discussion section,

⋮
⋮

We employed both idealised slope and Antarctic bed topography models to simulate ice sheet dynamics. The low geometric complexity of idealised slopes allows the comparative analyses of ice sheet dynamics to isolate the sole effects of ice viscosity and friction. The idealised slope model provides a controlled geometry to systematically investigate the effect of different fluidity parameter ($A$) on ice flow velocity. The idealised slope has previously been adopted to improve specific physical processes, such as marine ice sheet modelling and ice sheet instability (Favier et al., 2012; Pattyn et al., 2012). By demonstrating the effects of $A$ value in simplified conditions, we can address the effect of geometric complexity for Antarctic bed topography. We found that the wider range of $A$ values was derived from the idealised slope models, compared to the Antarctic bed topography models. However, when we used the Antarctic bed topography models, which incorporate complex bed topography, the effective range of $A$ values is narrower. This means that the $A$ value broadly determines the viscous ice flow, and the bed geometry and friction further modulate the velocity.

⋮
⋮

*Finally, the authors discuss in the Discussion section the effect of the sliding exponent m, which is a poorly constrained parameter and could also affect Antarctic velocities. However, the friction coefficient C is not discussed in this section and also has an effect on ice velocities. The authors should discuss how they chose the value of C, particularly in the Antarctic models, and whether this affects their results. In theory, one might imagine that you could obtain the velocities with one combination of A and C values and obtain the same velocities with a very different A and an appropriately tuned C.*

**Authors' reply:**

We appreciate the reviewer's insightful comments on the importance of the friction coefficient C in influencing ice velocities. In our study, the choice of C value was guided by the published literature. For example, Brondex et al. (2017) investigated the sensitivity of the friction law to grounding line dynamics. The numerical model setup was identical to MISMIP with spatially and temporally constant parameter values (e.g., A, m, and C values). The contribution of the grounding line and ice velocity variation to the ice flow model was compared for four different friction laws (Weertman, Schoof, Tsai, and Budd friction laws). The results highlighted the importance of selecting the appropriate friction law to improve the accuracy of ice flow models and predictions. Another example is Favier and Pattyn (2015), where the constant C values are used.

Contour plots of $\tau_b$ given in the plane ($N$, $u_b$) are represented in Fig. 1 for the four friction laws (Eqns (1)–(4)) using $m = 1/3$, $q = 1$, $C_W = C_B = C_S = 7.624 \times 10^6$ S.I. and $f = C_{max} = 0.5$; with this choice the four laws give the same $\tau_b$ for the value $N = 1$ MPa (highlighted by the black vertical dashed lines in Figs 1a–d). By definition, $\tau_b$ is independent

**Sensitivity of grounding line dynamics to the choice of the friction law (Brondex et al., 2017)**

**Table 1.** Model and Bed Topography Parameters

| Parameter | Symbol | Value | Unit |
|---|---|---|---|
| Flow parameter | $A$ | $3.10^{-25}$ | $Pa^{-3}\,s^{-1}$ |
| Seconds per year | | 31,536,000 | $s\,a^{-1}$ |
| Accumulation rate | $a_s$ | 0.3 | $m\,a^{-1}$ |
| Basal melting/accretion | $a_b$ | 0 | $m\,a^{-1}$ |
| Glen's exponent | $n$ | 3 | |
| Bed friction parameter | $C$ | $7.624 \times 10^6$ | $Pa\,m^{-1/3}\,s^{1/3}$ |
| Bed friction exponent | $m$ | 1/3 | |
| Sea density | $\rho_w$ | 1,000 | $kg\,m^{-3}$ |
| Ice density | $\rho_i$ | 900 | $kg\,m^{-3}$ |
| Gravity | $g$ | 9.8 | $kg\,m^{-3}$ |
| Domain length | $L$ | 800 | km |
| Domain width | $W$ | 256 | km |
| Maximum refinement | | 0.5 | km |
| Bed parameter | $\alpha$ | 1.9 | |
| Bed parameter | $\beta$ | 0.75 | |

to vertical shearing is neglected because of its effect of significantly reducing the time step. However, with respect to grounding line migration, this approximation in combination with a subkilometer grid spacing across the grounding line gives results that are in accord with full Stokes modeling [*Pattyn et al.*, 2013]. Furthermore, grounding line dynamics are better represented than in conventional SSA models or *Pollard and DeConto*, 2012-type [2012] parameterizations [*Pattyn and Durand*, 2013].

Besides the physical basis of the model, a subkilometric spatial resolution is a necessary condition to guarantee grounding line migration [*Pattyn et al.*, 2013; *Pattyn and Durand*, 2013]; therefore, the resolution ranges between 500 m at the grounding line and 4 km at the ice sheet divide and the calving front. Ice rheology is controlled by the Glen's flow law, and the interaction between the bed and the ice bottom surface by a Weertman-type nonlinear friction law [*Weertman*, 1957]. All model parameters are listed in Table 1.

**Antarctic ice rise formation, evolution, and stability (Favier and Pattyn, 2015)**

| Description | Symbols | Values [unit] and equations |
|---|---|---|
| Friction coefficient | $C_w$ | $7.624 \times 10^6$ $[Pa \cdot m^{-1/3} \cdot s^{1/3}]$ |
| Sliding exponent | $m$ | 1/3, 1/5, 1/7, and 1/9 [-] |

**Our study**

Basal drag using the Weertman friction law is one of the important factors controlling ice flow. Many previous studies conducted sensitivity tests (e.g., Gillet-Chaulet et al., 2016, Gilbert et al., 2023) of the friction law parameters (e.g., friction coefficient C and friction exponent m) for ice sheet dynamics. Gilbert et al. (2023) and Gillet-Chaulet et al. (2016) suggest that these friction-related parameters, particularly the friction exponent (m), can lead to uncertainty in modelling ice sheet dynamics. Gilbert et al. (2023) suggested that changes in ice sheet length are mainly driven by the glacier velocity in response to changes in ice thickness, which is controlled by the non-linearities in the friction law (exponent m) and Glen's law (exponent n). Assuming that the averaged ice viscosity and friction coefficient (C value) are constant over time and that the Glen's law exponent $n = 3$, the transient change in ice length is essentially controlled by the friction exponent m.

Gilbert et al. (2023) also show that the m value has a greater impact on the accuracy of the model predictions. The RMSE decreases from ~22 m/yr to ~12 m/yr (~100% decrease; see red arrow in the figure below) as the m values decrease from 5 to 3. Moreover, as the m value increases from 1 to 3, the RMSE decreases from 18.5 m/yr to ~12 m/yr (30% decrease; see blue arrow in the figure below).

[Figure]

**Figure 1.** (a, b) Observed velocity and thickness changes between September 2003 and 2018. (c, d, e) Modeled velocity change for different values of *m*. (f) Root mean squared error (RMSE) between observed and modeled velocity changes as a function of *m*.

Inferring the Basal Friction Law From Long Term Changes of Glacier Length, Thickness and Velocity on an Alpine Glacier (Gilbert et al., 2023)

Gillet-Chaulet et al. (2016) compared the effect of C value on the fit to observations (i.e., ice velocity). They tested both models with spatio-temporally varying C value and with constant C value. When C value is varied in time and space, the RMSE between model and observation was 18.9 m/yr. However, when the C value was varied for each velocity dataset (using inversion), the RMSE improved by only 10% to 16.5 m/yr. Although this difference suggests that changing the C value has some effect on the model predictions, the m value is a much more important parameter than the C value in modelling ice velocities. The m value determines the non-linearity (such as velocity-strengthening or velocity-weakening) between frictional traction and ice velocity along the bed, which is directly related to ice velocity. In contrast, the C value mainly represents the physical state of the bed (e.g. changes in bed topography), which is important but not as influential as the m value. We quoted a paragraph from Gillet-Chaulet et al. (2016).

**5. Conclusion**

With the assumption that the basal slipperiness coefficient, C, has been constant in time between 1996 and 2010, we have been able to reproduce most of the observed velocity changes in PIG with the basal friction law that is commonly used in ice-sheet models. The best match is obtained for a constant value of the stress exponent *m* = 20 with a RMSE of 18.9 m/a, while with a value of C that can vary with time (i.e., unique spatial fields for each velocity data set) the minimum improves only slightly to 16.5 m/a. Differences in RMSE are very

Assimilation of surface velocities acquired between 1996 and 2010 to constrain the form of the basal friction law under Pine Island Glacier (Gillet-Chaulet et al., 2016)

Following the approach of previous studies, we also assumed fixed values for the other parameters (e.g., C value and m) to isolate the effect of changes in the A value. As this is very similar to the objective of the study by Brondex et al. (2017), we assumed the same values for the friction law parameters (C and m), which are essential for our study. Furthermore, we investigated the effect of different m values of 1/3, 1/5, 1/7, and 1/9 on the ice velocity for the idealised slope model and the Antarctic bed topography model (please see the figure below). We also include the following figures in the Supplementary Material as follows.

[Figure]

Figure S4

We found that the variations in ice velocity depending on m values were not significant, thus increasing the confidence in the parameters obtained. Therefore, we have tested the important parameters n and m, which are mainly used in ice sheet dynamics, as constant values in time and space. This allows the calculated ice velocity change to be accurately constrained by isolating the effect of the A value. The friction law was set by adopting conventional C and m values used in studies where numerical simulations were generally performed using the Weertman friction law. The same basic friction coefficient (C) and exponent (m) were used as in the Marine Ice Sheet Instability (MISMIP) study by Favier and Pattyn (2015). Please see the tables in our manuscript and Favier and Pattyn (2015).

By thoroughly reviewing studies that use the Weertman Friction Law, we investigated the various friction coefficient (C) values employed when m = 1/3. It was confirmed that different studies adopted different C values. For example, Jong et al. (2018) used $C = 3.812 \times 10^6 \; \mathrm{Pa \cdot m^{-1/3} \cdot s^{1/3}}$ and m = 1/3, while Favier et al. (2012) performed a numerical simulation using the value $C = 10^7 \; \mathrm{Pa \cdot m^{-1/3} \cdot s^{1/3}}$ and m = 1/3. We quantified this by testing three cases with $C = 3.812 \times 10^6$ (Jong et al., 2018), $7.624 \times 10^6$ (our study), and $10^7$ (Favier et al., 2012) $\mathrm{Pa \cdot m^{-1/3} \cdot s^{1/3}}$.

[Figure]

Quoted from Jong et al. (2018)          Our study          Quoted from Favier et al. (2012)

We plotted the averaged ice velocity as C varied for all Antarctic models with $A = 0.02 \times$, $4.0 \times$, and $64.0 \times 10^{-32} \mathrm{Pa^{-4} s^{-1}}$ (see figure below). Compared to the variation of averaged ice velocity with the value of the fluidity parameter (A) of Glen's law, the variation with the value of C was relatively small (less than 10%). We added the result for the C value to the supplementary material.

[Figure]

Figure S5. The calculated variation in ice flow velocity on the Antarctic ice shelf as a function of the sliding coefficient (C). The commonly used value of C is $7.624 \times 10^6$ Pa · $m^{-1/3} \cdot s^{1/3}$. We compared the ice velocity changes for values of C = $10^7$ Pa · $m^{-1/3} \cdot s^{1/3}$ and $3.81 \times 10^6$ Pa · $m^{-1/3} \cdot s^{1/3}$. (a), (b), and (c) correspond to models with $A =$ 0.02, 4.0, and $64.0 \times 10^{-32}$ Pa$^{-4}$ s$^{-1}$, respectively.

*Results: The range of A values ultimately constrained by these simulations is quite narrow, far more narrow than the expected spatial variation in A due to heating, fabric (which itself can affect A by 0.5-1 order of magnitude), damage (which can affect A by many orders of magnitude, in theory), water content, etc. I believe this result would be different if the authors used an inverse method, but if the authors choose not to, it's important to put this range into context – in particular, that it is a range of average values, and in dynamical regions of ice sheets, this value may be significantly higher or (possibly) lower due to material and physical properties.*

**Authors' reply:**

As Reviewer #2 points out, the range of A values obtained by our simulations (one order of magnitude) is much narrower than expected in nature. We found that the range of A values was narrowly constrained because the ice velocity on the Shackleton Ice Shelf is too slow compared to other ice shelves. This was also pointed out by another reviewer. To make the range of A values more general, we have included modelled ice velocities from the Cook Ice Shelf and Pine Island Glacier in our analysis in the revised manuscript. Despite the addition of these two ice shelves, the constrained range of A values has not changed from that of the original draft.

If we remove the ice velocities of the Shackleton Ice Shelf from our analysis, the range is extended to an order of three (compare left and right figures below). Although the order of the A value range is increased from one to three, an order of three is still meaningful. Typically, for n = 3, the range of A values used in modelling ice sheet dynamics has an order of three.

We carefully considered removing the Shackleton Ice Shelf as an outlier. In the end, we decided that we could not remove the Shackleton Ice Shelf because it is obviously an Antarctic glacier. However, we argued in the discussion section that if we removed Shackleton Ice Shelf, the order of the A value range would be three.

[Figure]

| Includes all ice shelves | Excludes the Shackleton Ice Shelf, and adds the Cook Ice Shelf and Pine Island Glacier |
|---|---|

We acknowledge that the range of A values for our study is quite narrow, given the effects of temperature change, ice crystal structure, damage, and water content on A values. Therefore, we have included this limitation of our study in the Discussion section: we have not included changes in A value due to other physical factors (e.g. temperature, ice crystal structure, and damage). However, the aim of our study was to refine the range of A value as much as possible (when n = 4). The refined A values can then be used as initial values for the inversion method (inverting the A value distribution from the observed velocity) or as the prefactor $A_0$ for the temperature dependence relationship ($A = A_0 \exp\left(-\frac{E_a}{RT}\right)$). We thank the reviewer again for the valuable feedback.

**In the Discussion section,**
   Millstein et al. (2022) showed that a value of $n = 4.1 \pm 0.4$ better approximates viscous ice flow than the commonly used $n = 3$ in fast-flowing and highly stressed ice

shelves (e.g., Ronne and Ross Ice Shelf). Higher values of n increase the sensitivity of viscosity to changes in stress and temperature, which can lead to significant variations in numerical simulations. Therefore, when calculating ice flow velocity with $n = 4$, sensitivity analysis of other parameters in Glen's law, such as the A value. We have performed numerical simulations using the fluidity parameter ($A$) as a constant value in ice dynamics models. However, the various physical factors (e.g. temperature, ice crystal fabric, and ice grain size) affect the value of the fluidity parameter. The fluidity parameter ($A$) is influenced by temperature, ice structure, grain size, damage and moisture content, which in turn affect ice flow dynamics. Cuffey and Paterson (2010) suggested that temperature changes in the range -10 °C to 0 °C have a significant effect on the A value, by influencing the viscosity and deformability of the ice. Furthermore, ice crystal fabric and grain size can determine the response of the glacier to applied stresses (Goldsby and Kohlstedt et al., 2001). Structural damage within the glacier, such as cracks and microdefects, can significantly reduce deformation resistance, enhancing glacier flow (Duddu and Waisman, 2012). The water content of the glacier ice also acts as a lubricant, reducing internal friction and increasing flow, particularly promoting basal sliding. However, accurate prediction of ice rheology parameters is challenging due to multiple interrelated factors such as water, temperature, ice fabric, particle size, and damage. Based on the narrow range of $A$ values refined in this study, we expect to derive $A$ values that more accurately represent the behaviour such as ice temperature and ice velocity variations.

⋮
⋮

*Other Items:*

*Citations in lines 30-33 could be adjusted. Citations for n=3 could include Jezek et al. 1985, Martin and Sanderson 1980, Paterson et al. 1983, and the original Glen papers (Glen 1955). I also believe that Behn et al. 2021 was not primarily using geodetic data, they were applying ice core and laboratory data along with models. Further, the statement that n=3 reproduces surface ice velocity does not seem to be supported by the citations, as I believe most of the citations were looking at field observations, both at the surface and at depth. If any of this is not correct, please feel free to ignore, but it may be worth double-checking these citations.*

**Authors' reply:**

We deeply appreciate the valuable feedback from the reviewer. We have added Jezek et al. (1985), Martin and Sanderson (1980), Paterson et al. (1983) and Glen (1955) to the Introduction section for the exponent of Glen's law n = 3. In response to the reviewer's comment that Behn et al. (2021) did not primarily use geodetic data but rather applied the model in combination with ice core and laboratory data, we have corrected this part as follows.

**In the Introduction section,**
The value of $n$ approximately 3 has been used, which effectively reproduces the surface ice velocities in Antarctica (Glen, 1955; Martin and Sanderson, 1980; Paterson et al., 1983;

Jezek et al., 1985; Pattyn et al, 2012), such as the Amery Ice Shelf (Thomas, 1973; Hamley et al., 1985) by laboratory experiment.

*In the paragraph starting at line 34, I believe Millstein et al. 2022 and Bons et al. 2018 should be cited here, since they are both observational studies suggesting that n varies but tends to n=4 in their study regions. I know that the authors cite both these studies in the next paragraph about A, but as both of these studies are primarily about n, they fit into this paragraph as well.*

**Authors' reply:**

Following the reviewer's suggestion, we have added two more references of Millstein et al. (2022) and Bons et al. (2018), where the authors' opinion is that n tends to approach 4. The reviewer's suggestion ensures that we fairly cite previous studies with n = 3 and studies with n = 4.

> **In the Introduction section,**
> A detailed depth analysis of deep ice cores from Antarctica and Greenland revealed large variations in grain shape, grain size, and anisotropy within the ice structure (Budd and Jacka, 1989; Cuffey et al., 2000; Bons et al., 2018; Millstein et al., 2022), indicating that stress exponent values other than three can be considered when fitting ice viscosity.

*L41: The fluidity parameter A is also affected by fabric, damage, impurities, among others*

**Authors' reply:**

We have followed the reviewer's suggestions.

> **In Introduction section,**
> **Line #41:** The $A$ is modulated by the temperature,  fabric, grain size, damage, and impurities ( Hruby et al., 2020; Adams, 2021).

*Citations*

*Larour (2005), Rheology of the Ronne Ice Shelf, Antarctica, inferred from satellite radar inferometry data using an inverse control method, Geophysical Research Letters (32)5, doi: 10.1029/2004GL021693*

*Jezek, et al. (1985), Rheology of Glacier Ice, Science (227)4692, doi: 10.1126/science.227.4692.1335*

*Paterson (1983), Deformation within polar ice sheets: An analysis of the Byrd Station and Camp Century borehole-tilting measurements, Cold Regions Science and Technology, (8)2, doi: 10.1016/0165-232X(83)90007-1*

*Glen (1955), The creep of polycrystalline ice, Proceedings of the Royal Society A: Mathematical, Physical, and Engineering Sciences*

*L45: what is the "previous ice sheet model"?*

**Authors' reply:**

In the original manuscript, "the previous ice sheet model" refers to Durand et al. (2009), in which the temporal evolution of A drives a hysteretic behaviour of the grounding line. We have revised the sentence to avoid the confusion with "the previous ice sheet model".
* * *
**In the Introduction section,**

Furthermore,  Durand et al. (2009) showed that the temporal evolution of $A$ yields the hysteretic behaviour of the grounding line, implying the need to constrain the range of $A$.
* * *
*L46: The sentence "The values of A recent inference…argues for n of approximately four" I had trouble understanding*

**Authors' reply:**

Millstein et al. (2022) investigated n values for the Antarctic ice shelf. Millstein et al. (2022) showed that a value of n = 4.1±0.4 better approximates viscous ice flow than the commonly used n = 3. Thus, ice sheet dynamics models with n = 4 has the potential to significantly increase the sensitivity of ice sheet mass loss to ongoing climate change, compared to n = 3. Furthermore, the higher values of n increase the sensitivity of viscosity to stress variations, which can lead to large errors in numerical simulations. Therefore, sensitivity analyses of other parameters (e.g., the A value in our study) are required when calculating ice flow for n = 4. We quotd a core sentence from Millstein et al. (2022).
* * *
The log–log plots between strain rate and deviatoric stress shown in Fig. 2 exhibit linear trends that are consistent with a power-law relation. These results provide strong evidence that, for a suitable choice of $n$, Glen's Flow Law is a viable approximation of the viscous flow of Antarctic ice shelves and, as discussed later, likely other dynamic regions of Antarctica. Critically, we find $n \approx 4$ in the fast-flowing, extensional regions of Antarctic ice shelves. This result is consistent with other evidence for a

Ice viscosity is more sensitive to stress than commonly assumed (Millstein et al., 2022)
* * *
We have corrected the sentence to reduce confusion.
* * *
**In the Introduction section,**

**Line #46:** The values of $A$ recent inference of the stress exponent ( Millstein et al., 2022) argues for n of approximately four better approximates viscous ice flow than the commonly used $n = 3$, especially for fast-flowing
* * *
and highly stressed ice shelf (e.g., Ross Ice Shelf).

*L139: 2000 m/yr velocities are not higher than the maximum value in Antarctica (Pine Island Glacier has velocities at or near 4000 m/yr), but it is on the high side for the average of ice shelves*

**Authors' reply**:

We thank the reviewer for this comment. We compared the modelled ice velocity with the range of ice velocities from a subset of NASA's Making Earth System Data Records for Use in Research Environments (MEaSUREs) from 2007 to 2009, covering the whole of Antarctica. Our study focused specifically on the ice velocities of the ice shelf at the ice front. We set the model sites to include the ice shelves, which also have a high flow velocity. We excluded the velocity calculated when $A = 10^{-29}$ $Pa^{-4}s^{-1}$ because it was much higher than the average ice velocity on the Antarctic ice shelf from 2007 to 2009.
* * *
**In Results section,**
**Line #139:** For $A = 10^{-29}$ $Pa^{-4}s^{-1}$, the ice velocity was ~2000 m/year, which was higher than the maximum value in Antarctica ice velocities from 2007 to 2009 that we adopted.
* * *
*Fig 4: the yellow and orange lines are a bit hard to see*

*In general, I would recommend italicizing A and n, or using the Latex math environment, to distinguish them from the prose*

**Author's reply:**

To improve the readability of the yellow and orange lines in Figure 4, we increased the colour contrast.

[Figure]

Original Figure 4         Revised Figure 4

Using the LaTeX mathematical setting, including the support of italics, we were able to clearly distinguish mathematical variables such as A and n.
* * *
**In the Abstract section,**

···The suggested range of the fluidity parameter ( $A$) for  $n = 4$ is of the order of six (i.e., $10^{-35}$ to $10^{-29}$ Pa$^{-4}$s$^{-1}$), leading to a significant uncertainty in ice velocity than when  $n = 3$.

**In Introduction section,**

···The value of  $A$ derived from a laboratory ice deformation experiment with  $n = 3$ ranged from ···

**In Method section,**

···effect of fluidity parameters when  $n = 4$, ···

**In the Discussion section,**

···Recent ice sheet dynamics models have argued that Glen's law with  $n$ (power-law stress exponent) = 4, instead of  $n = 3$···

**In the Conclusions section,**

···with  $n = 4$
* * *
We thank Reviewer once again for the valuable time and consideration.

Sincerely,

Sujeong Lim and Prof. Byung-Dal So

**References for this response letter**

Aster, R. C., Borchers, B., and Clifford, H. T.: Parameter estimation and inverse problems, Elsevier, Amsterdam, the Netherlands, 2005.

Behn, M. D., Goldsby, D. L., and Hirth, G.: The role of grain size evolution in the rheology of ice: implications for reconciling laboratory creep data and the Glen flow law, The Cryosphere, 15, 4589–4605, https://doi.org/10.5194/tc-15-4589-2021, 2021.

Bons, P. D., Kleiner, T., Llorens, M.-G., Prior, D. J., Sachau, T., Weikusat, I., and Jansen, D.: Greenland Ice Sheet: Higher nonlinearity of ice flow significantly reduces estimated basal motion, Geophys. Res. Lett., 45, 6542–6548, http://doi.org/10.1029/2018GL078356, 2018.

Brondex, J., Gagliardini, O., Gillet-Chaulet, F., and Durand, G.: Sensitivity of grounding line dynamics to the choice of the friction law, J. Glaciol., 63, 854–866, https://doi.org/10.1017/jog.2017.51, 2017.

Choi, Y., Seroussi, H., Morlighem, M., Schlegel, N. J., and Gardner, A.: Impact of time-dependent data assimilation on ice flow model initialization and projections: a case study of Kjer Glacier, Greenland, The Cryosphere, 17, 5499–5517, https://doi.org/10.5194/tc-17-5499-2023, 2023.

Durand, G., Gagliardini, O., de Fleurian, B., Zwinger, T., and Le Meur, E.: Marine ice sheet dynamics: Hysteresis and neutral equilibrium, J. Geophys. Res., 114, 1–10, https://doi.org/10.1029/2008JF001170, 2009.

Favier, L. and Pattyn, F.: Antarctic ice rise formation, evolution, and stability, Geophys. Res. Lett., 42, 4456–4463, https://doi.org/10.1002/2015GL064195, 2015.

Favier, L., Gagliardini, O., Durand, G., and Zwinger, T.: A three dimensional full Stokes model of the grounding line dynamics: effect of a pinning point beneath the ice shelf, The Cryosphere, 6, 101–112, http://doi.org/10.5194/tc-6-101-2012, 2012.

Gilbert, A., Gimbert, F., Gagliardini, O., and Vincent, C. : Inferring the basal friction law from long term changes of glacier length, thickness and velocity on an alpine glacier, Geophys. Res. Lett., 50, e2023GL104503. https://doi.org/10.1029/2023GL104503, 2023.

Gillet-Chaulet, F., Durand, G., Gagliardini, O., Mosbeux, C., Mouginot, J., Rémy, F., and Ritz, C.: Assimilation of surface velocities acquired between 1996 and 2010 to constrain the form of the basal friction law under Pine Island Glacier, Geophys. Res. Lett., 43, 10–311, 2016.

Gladstone, R. M., Xia, Y., and Moore, J.: Neutral equilibrium and forcing feedbacks in marine ice sheet modelling, The Cryosphere, 12, 3605–3615, https://doi.org/10.5194/tc-12-3605-2018, 2018.

Glen, J. W.: The creep of polycrystalline ice, Proc. Roy. Soc. Lond. A., 228, 519–538, https://doi.org/10.1098/rspa.1955.0066, 1955.

Jezek, K. C., Alley, R. B., and Thomas, R. H.: Rheology of glacier ice, Science, 227, 1335–1338, http://doi.org/10.1126/science.227.4692.1335, 1985.

Jong, L. M., Gladstone, R. M., Galton-Fenzi, B. K., and King, M. A.: Simulated dynamic regrounding during marine ice sheet retreat, The Cryosphere, 12, 2425–2436, https://doi.org/10.5194/tc-12-2425-2018, 2018.

Martin, P. J. and Sanderson, T. J. O.: Morphology and Dynamics of Ice Rises, J. Glaciol., 25, 33–46, https://doi.org/10.3189/S0022143000010261, 1980.

McArthur, K., McCormack, F. S., and Dow, C. F.: Basal conditions of Denman Glacier from glacier hydrology and ice dynamics modeling, The Cryosphere, 17, 4705–4727, https://doi.org/10.5194/tc-17-4705-2023, 2023.

Millstein, J. D., Minchew, B. M., and Pegler, S. S.: Ice viscosity is more sensitive to stress than commonly assumed, Communications Earth & Environment, 3, 57, https://doi.org/10.1038/s43247-022-00385-x, 2022.

Paterson, W. S. B.: Deformation within polar ice sheets: An analysis of the Byrd Station and Camp Century borehole-tilting measurements, Cold Reg. Sci. Tech., 8, 165–179, https://doi.org/10.1016/0165-232X(83)90007-1, 1983.

Pattyn, F., Schoof, C., Perichon, L., Hindmarsh, R. C. A., Bueler, E., de Fleurian, B., Durand, G., Gagliardini, O., Gladstone, R., Goldberg, D., Gudmundsson, G. H., Huybrechts, P., Lee, V., Nick, F. M., Payne, A. J., Pollard, D., Rybak, O., Saito, F., and Vieli, A.: Results of the Marine Ice Sheet Model Intercomparison Project, MISMIP, The Cryosphere, 6, 573–588, https://doi.org/10.5194/tc-6-573-2012, 2012.

Wolovick, M., Humbert, A., Kleiner, T., and Rückamp, M.: Regularization and L-curves in ice sheet inverse models: a case study in the Filchner–Ronne catchment, The Cryosphere, 17, 5027–5060, https://doi.org/10.5194/tc-17-5027-2023, 2023.

---

## Author Comment (AC2)

**Response Letter to Reviewer #1**

Blue letters: revised and/or added text in the revised manuscript

 with strikethrough: deleted text in the revised manuscript

Text in *italic*: comments from the Reviewer #1

*General Comments*

*Based on recent work (Millstein et al., 2022; Ranganathan et al., 2021) there is increasing interest in using n=4 rather than n=3 in ice sheet modelling. This paper uses 2D full-Stokes simulations with n=4 to provide order-of-magnitude constraints on the value of the fluidity parameter when n=4. They reduce the range of A from 6 orders of magnitude to 1. Generally, I think the paper provides a useful contribution to the community and should be published after the inclusion of a few more clarifying figures and paragraphs.*

**Authors' reply:**

We sincerely appreciate Reviewer #1's cheerful and insightful comments. Our manuscript has been improved based on Reviewer #1's suggestions. We have included the responses and materials (e.g., figures and references) in this response letter.

*Originality: The paper performs simple 2D simulations of regions of ice sheets. The novelty comes from using the value of n = 4 and exploring the range of A values in the rheology which give reasonable results. The main contribution is new constraints on the value of A to use when using n=4. This will be useful to ice sheet modellers interested in using this value of n, of which there is increasing interest in.*

**Authors' reply:**

As Reviewer #1 correctly points out, we designed this study to narrow the range of A values for n = 4 using different Antarctic ice sheets and idealised slope models. We believe that our work can be a useful reference for the ice sheet modelling community in selecting A values when n = 4.

*Scientific Quality: The authors use 2D full-stokes simulations to constrain appropriate value of A which give predictions which match observed velocities. The use of full-Stokes is good, and I am also pleased the authors demonstrate that the results are robust to a range of sliding exponents.*

**Authors' reply:**

We appreciate the constructive feedback and positive evaluation of our work. To meet the high standards of numerical modelling in the ice sheet dynamics community, we recognised the need to test many parameters other than the A value, such as the sliding exponent. We have also done our best to perform the additional tests suggested by Reviewer #1, including 3D modelling of the idealised slope models.

*One concern I have about the final range of A values given is that it may over-constrain A. The final range of A values stated, $t4.0o\ 16 \times 10^{-32}\ Pa^{-4}s^{-1}$, is found by simply combining the A values which give reasonable results for each case studied. One could imagine that if more ice shelves were included in this study, this method would give no A value which gave reasonable results for all. This is not a criticism of the work, which I think is valid, but just the calculation procedure used at the end.*

**Authors' reply:**

We appreciate Reviewer #1's concern about the possibility of over-restricting the range of A values. We also recognise that our results should be applicable to a wider region of Antarctica. Furthermore, our result should not be restricted to the limited cases of specific ice shelves. To address the concerns of Reviewer #1, we have performed additional analyses to address the limitations of our approach for determining the range of A values, specifically, 4.0 to $16 \times 10^{-32}\ Pa^{-4}s^{-1}$. To address the concerns of Reviewer #1, we have extended our study to include additional numerical experiments on two other Antarctic ice shelves, the Cook Ice Shelf and Pine Island Glacier. These two ice shelves were chosen because of their different physical parameters (e.g., bed topography, surface elevation, ice thickness, and shelf length) and ice deformation.

[Figure]

Revised Figure 1.
Antarctic bed topography models with Cook Ice Shelf and Pine Island Glacier

The figures below show the numerical results for Cook Ice Shelf and Pine Island Glacier. The experiments were performed with the same range of A values ($10^{-35}$ to $10^{-29}$ Pa$^{-4}$s$^{-1}$), as used in the original manuscript. We found that the Pine and Cook glaciers yielded ice flow velocities corresponding to the average Antarctic ice velocities (see the grey shaded zone in the figures below) in the A value range of $10^{-35}$ to $63.0 \times 10^{-32}$ Pa$^{-4}$s$^{-1}$ and $10^{-35}$ to $251.0 \times 10^{-32}$ Pa$^{-4}$s$^{-1}$, respectively. The final range of A values, including Cook Ice Shelf and Pine Island Glacier, is the same as the range of A values presented in this study.

[Figure]

Variation of ice flow velocity with A-values on the Cook Ice Shelf and Pine Island Glacier

*The Shackelton Ice shelf in particular is an outlier in terms of the appropriate range of A values. It would be useful to explore in the discussion what the suggested range of A values would be without this case and have some statistical analysis and a final summarizing figure showing the suggested A values for each case examined.*

**Authors' reply:**

We appreciate the reviewer's insightful feedback and suggestions on the A value analyses, particularly for the Shackleton Ice Shelf, which has extremely slow ice flow compared to the other ice shelves. In this revision, we report two different ranges of A values from the model sets: one including the Shackleton Ice Shelf and one excluding it. The comparison suggested that removing the Shackleton Ice Shelf from the model set would increase the range of A values. When the Shackleton was included (as in the original manuscript), the range of A values was $4.0 \times$ to $16.0 \times 10^{-32}$ Pa$^{-4}$s$^{-1}$ (an order of one). However, when the Shackleton was removed (for the revised manuscript), the range of A was extended, from $0.06 \times$ to $16.0 \times 10^{-32}$ Pa$^{-4}$s$^{-1}$ (an order of three). We also confirmed that including additional Antarctic ice shelves (i.e., Cook Ice Shelf and Pine Island Glacier) to this analysis resulted in the same range (i.e., $4.0 \times$ to $16.0 \times 10^{-32}$ Pa$^{-4}$s$^{-1}$) as in the original manuscript. We believe that this additional experiment provides a stronger generalisation for this range of A values.

[Figure]

| Includes all ice shelves (in the original draft) | Excludes the Shackleton Ice Shelf, and adds the Cook Ice Shelf and Pine Island Glacier |
|---|---|

We admit that the A values derived from including the Shackleton Ice Shelf further narrow the range of A values. As stated above, the model set without the Shackleton resulted in an increased range of A values (an order of three). Although the order of the A value range is increased from one to three, an order of three is still meaningful. Typically, for n = 3, the range of A values used in modelling ice sheet dynamics has an order of three (Millstein et al., 2022).

However, we want to recall that the aim of our study was to improve the understanding of the complex rheological behaviour when n = 4. To limit the range of A values in order to minimise the errors in the numerical simulations caused by the large range of A values, using both idealised and Antarctic bed topography models, which are widely used in ice sheet dynamics in general.

Although we seriously considered removing the Shackleton results from the manuscript because Shackleton can be considered as an outlier, we ultimately decided in the end to report "an order of one" so that the A values would have a wider application across the Antarctic shelves, in line with the aims of our study.

We have also noted in the Discussion section that in the absence of the Shackleton, the A value range is of the order of three. Furthermore, we emphasised that this order of three (i.e., $0.06 \times$ to $16.0 \times 10^{-32}$ Pa$^{-4}$s$^{-1}$) is significantly narrower than the experimentally obtained A value range, which is an order of six ($10^{-35}$ to $10^{-29}$ Pa$^{-4}$s$^{-1}$).

*There is also likely sufficient unaccounted for physics implicit in A, such as temperature variation, fabric induced anisotropy, grain size etc. which would give an order of magnitude variation at least in the true value. This should also be mentioned in the discussion.*

**Authors' reply:**

We very much appreciate the reviewer's comment about mechanics, which may not have been considered when setting the A value range in our study. Indeed, factors such as temperature change, fabric-induced anisotropy, and grain size are indeed very important in controlling ice viscosity (Gillet-Chaulet et al., 2006; Hruby et al., 2020; Adams et al., 2021). We admit that in our original manuscript we didn't sufficiently discuss the complicated interaction of different physical factors on the A value. In this response letter, we have briefly reviewed the literature to date on the effects of temperature variation, fabric-induced anisotropy, grain size, damage, and water content on the mechanical properties of ice, which in turn affect the flow dynamics of ice.

**Review of possible factors affecting A values**

**1) The effect of temperature on the A value**

The influence of temperature on the A value in Glen's law, which is the prefactor to the $n^{th}$ power of the deviatoric stress, is a critical aspect of understanding glacier dynamics. Viscosity, in turn, determines the resistance of the ice to deformation, a key factor in glacier flow and creep. Warmer and less viscous ice can exhibit a more fluid-like behaviour compared to that of colder and more rigid ice. The temperature dependence of ice flow can be quantified by a change in the A value. A glaciology textbook by Cuffey and Paterson (2010) summarised the exponential relationship between temperature and the A value, highlighting the profound sensitivity of ice to temperature variations. Even within a narrow temperature range (-10°C to 0°C), the variation in the A value is large. The figure below is a plot of the temperature dependence of A from Jasen et al., 2005. A has relatively low values when the ice temperature is low. This means that colder ice is more difficult to flow. In particular, it can be seen that the value of A increases rapidly as the temperature rises near the melting point of the ice.

[Figure]

Model experiments on large tabular iceberg evolution: ablation and strain thinning (Jansen et al., 2005)

Typically, the A value is parameterised with an activation energy for ice creep, which

quantifies the energy required to initiate deformation. The exponential increase in the A value with temperature reflects the decrease in the effective energy barrier to ice deformation, allowing more pronounced flow as temperatures approach the melting point.

**2) The effect of ice fabric and grain size on the A value**

The intricate relationship between crystal orientation, texture, and flow dynamics is a subject of great interest in glaciology. Ice-crystal fabric (Lilien et al., 2021) refers to the orientation and arrangement of ice crystals in glaciers, which plays an important role in the internal flow and deformation processes of glaciers. Ice-crystal fabric provides clues to how glaciers move and deform by describing how ice crystals are arranged according to the direction of glacier movement. The role of fabric in determining the A value is also significant, although it is difficult to quantify due to its dependence on specific fabric types, as well as the deformation history of the ice (Hudleston et al., 2015). Azuma (1994) suggests that the presence of a strong single maximum fabric can increase the deformation rate compared to isotropic ice under similar stress regimes. Strongly aligned crystal fabrics promote deformation by facilitating sliding along specific crystal planes, thereby increasing A. However, the effect of fabrics on glacier flow is complex, depending on both the deformation history and temperature conditions (Wilson and Peternell, 2012). This complexity suggests that the glaciers with similar climatic temperatures and different flow histories may exhibit different flow behaviour. Previous studies of ice deformation experiments (Goldsby and Kohlstedt et al., 2001; Behn et al., 2021) showed that grain size has a significant effect on strain rate, with strain rate increasing as grain size decreases. The quantitative relationship between grain size and the A value has been investigated in several studies.

[Figure]

**Figure 3.** Effective stress vs. grain size at **(a)** 240 K and **(b)** 265 K calculated for a shear zone of fixed width using the wattmeter. Dark and light blue symbols correspond to the steady-state grain size predicted from a single model simulation at a given strain rate. Dashed red lines show location of the piezometer (Jacka and Li, 1994). Model results are overlain on a deformation mechanism map for ice calculated at the appropriate temperature using the flow law parameters from Goldsby and Kohlstedt (2001). Background contours correspond to strain rate; the thick black line indicates the boundary between GBS-limited creep (upper left) and dislocation creep (lower right). Under these conditions the location of the field boundary and piezometer are very similar.

The role of grain size evolution in the rheology of ice: implications for reconciling laboratory creep data and the Glen flow law (Behn et al., 2021)

**3) The effect of damage on the A value**

The concept of damage in ice sheet dynamics refers to the presence of structural irregularities (or weaknesses), such as cracks and microdefects, which significantly affect the strength of the ice (Duddu and Waisman, 2012). These imperfections in the ice matrix act as

focal points for strain concentration, facilitating deformation at relatively low applied stresses, compared to undamaged ice. From a physical perspective, the damaged zones within the ice matrix can act as conduits for deformation, allowing the ice to adapt and flow even under relatively low stress conditions (Stone et al., 1997). It would therefore be expected that ice with a high degree of damage would have a higher A value. The relationship between damage and the A value highlights the importance of considering structural integrity and damage mechanisms when modelling glacier flow. For example, the presence of extensive crevasse or frost damage can alter local flow dynamics, as well as the overall stability and evolution of the glacier system.

[Figure]

Experiments on the damage process in ice under compressive states of stress (Stone et al., 1997)

**4) The effect of water content on the A value**

The water content of glacier ice is a key factor that significantly influences the dynamics of ice flow, primarily through its effect on the internal friction of the ice (Brown et al., 2017; Adams et al., 2021). The presence of water in a glacier lubricates the space between ice crystals, reducing the overall viscosity of the ice mass.

The effect of water on glacier flow can be particularly noticeable at the base of a glacier. Basal sliding, the process by which a glacier slides over its bed, is greatly enhanced in the presence of water. This lubricating effect accelerates glacier movement, which adds to the complexity of predicting glacier dynamics due to the non-linear response of flow rates to changes in water content. Adams et al. (2021) argued that, below a water content threshold of 0.6%, ice viscosity decreases significantly with increasing water content. However, above the 0.6% threshold, the viscosity dependence on water content becomes negligible, with the viscosity remaining almost constant. This implies a transition in the ice creep mechanism that depends on the water contents. Thus, the relationship between water content and A value should be parameterised.

[Figure]

FIGURE 6 | Effective ice viscosity as a function of water content, as measured in the 11 experiments to peak stresses and one tertiary-creep experiment of this study, compared with the tertiary-creep results of Duval (1977). Duval's rotary experiments were conducted at a constant shear stress of 0.29 MPa.

Softening of temperate ice by interstitial water (Adams et al., 2021)

Ice rheological parameters, such as A value, are difficult to predict accurately in environments with multiple interrelated factors (e.g. water content, temperature, ice fabric, grain size, and damage). In our study, we attempted to quantify the effect of the A value, which is highly dependent on environmental factors, especially temperature. Many previous studies have assumed a constant value of A (over space and time) for numerical simulations without considering other environmental factors (Favier et al., 2012), especially for simulations with n = 3.

Following the efforts of previous studies to isolate the effect of A value on ice dynamics, we have restricted the wide range of constant A values to a narrower range at n = 4. However, we fully acknowledge the need to consider the spatially and temporally variable distribution of A values. We have added a new discussion of various factors affecting A values, including water content, temperature, grain size and damage, in the Discussion section, with relevant references, as reviewed above.

**In the Discussion section,**

Millstein et al. (2022) showed that a value of $n = 4.1 \pm 0.4$ better approximates viscous ice flow than the commonly used $n = 3$ in fast-flowing and highly stressed ice shelves (e.g., Ronne and Ross Ice Shelf). Higher values of n increase the sensitivity of viscosity to changes in stress and temperature, which can lead to significant variations in numerical simulations. Therefore, when calculating ice flow velocity with $n = 4$, sensitivity analysis of other parameters in Glen's law, such as the $A$ value. We have performed numerical simulations using the fluidity parameter ($A$) as a constant value in ice dynamics models. However, the various physical factors (e.g. temperature, ice crystal fabric, and ice grain size) affect the value of the fluidity parameter. The fluidity parameter ($A$) is influenced by temperature, ice structure, grain size, damage and moisture content, which in turn affect ice flow dynamics. Cuffey and Paterson (2010) suggested that temperature changes in the range -10 °C to 0 °C have a significant effect on the $A$ value, by influencing the viscosity and deformability of the ice. Furthermore, ice crystal fabric and grain size can determine the response of the glacier to applied stresses (Goldsby and Kohlstedt et al., 2001). Structural damage within the glacier, such as cracks and microdefects, can significantly reduce

deformation resistance, enhancing glacier flow (Duddu and Waisman, 2012). The water content of the glacier ice also acts as a lubricant, reducing internal friction and increasing flow, particularly promoting basal sliding. However, accurate prediction of ice rheology parameters is challenging due to multiple interrelated factors such as water, temperature, ice fabric, particle size, and damage. Based on the narrow range of $A$ values refined in this study, we expect to derive $A$ values that more accurately represent the behaviour such as ice temperature and ice velocity variations.

$$\vdots$$
$$\vdots$$

In ice sheet dynamics modelling, the inversion method has been widely used to derive spatially and temporally variable rheological and frictional parameters, as highlighted in previous studies (Choi et al., 2023; Wolovick et al., 2023). In addition to improving the ability of models to fit observed ice velocities, such methods ultimately enhance the predictive capability of future ice behaviour. However, we focused on the use of a constant value of $A$ with a stress exponent ($n$) of 4. The constant value of $A$ provides a baseline value for the sensitivity of ice flow to rheological changes. The narrowed range of $A$ values in our study was achieved by minimising the error between the numerical model and observed Antarctic ice velocity, allowing the range of $A$ values to be used as initial values of temperature-dependent ice viscosity. We expect that this approach will provide insights for the development of refined model parameters.

*Significance: The paper will be useful to ice sheet modellers in general, and hopefully encourage more studies using n=4.*

*Presentation quality: The paper is generally well written and concise. The title provides a clear summary of the work, and the abstract communicates the main results, though I think it should be noted in the abstract that this is a 2D study.*

**Authors' reply:**

We are grateful for the reviewer's kind description of the title and abstract of our manuscript. We recognised the importance of explicitly mentioning the (two) dimensionality of the study. We have revised the abstract to state that the analysis was performed in a two-dimensional context. This change ensures that readers are immediately informed of the scope and methodological framework of our analysis, thereby avoiding any potential misunderstanding of the dimensionality of the study.

> **In the Abstract section,**
> ⋯Here, we refined $A$ to within one order, aligning with observed Antarctic ice velocities with two-dimensional simplified slope and Antarctic bed topography models. ⋯

·    *I would like to see at least one 3D simulation alongside the 2D cases for the idealised configuration explored here, to ensure that the same range of A values are found in 2D and the 3D cases. The limitation of 2 dimensions and how things may change in 3D should also be included in the discussion.*

**Authors' reply:**

We appreciate the reviewer's insightful suggestion to include the 3D simulation for the idealised models in our study. We have discussed the limitations of the 2D simulations and possible changes in the 3D simulation based on additional idealised 3D models.

Although two-dimensional (2D) models are useful for investigating plane-strain problems (very large in-plane width with low computational cost), 2D simulations only consider vertical and horizontal stresses. The 2D model has inherent limitations, particularly when modelling the complex dynamics of large glaciers or ice sheets, especially around the grounding line position, which is sensitive to stress changes in three-dimensional directions. Stress interaction in the in-plane direction can be important for the stability of ice sheet mechanisms. Although three-dimensional (3D) modelling leads to a significant increase in computational effort (in terms of cost and coding), 3D models can more accurately reflect the complex internal structure and dynamics of the entire ice sheet. In particular, changes in bed topography and ice flow along the in-plane width can have a significant effect on the behaviour of the ice sheet. These changes can be better captured in a three-dimensional model. In addition, detailed changes in stress distribution, including the interaction of lateral and vertical stresses, can be captured in the 3D model, which can have a dominant effect on the response of the ice sheet to different conditions. Through simple testing with an idealised 3D slope model, we can improve the predictive capability of our model for ice sheet response, which can ensure detailed analyses of ice strength.

We constructed an idealised 3D slope model to address the reviewer's concerns. The bottom (frictional boundary condition), top (free surface), back (no slip) and front (water pressure) boundary conditions were the same as for the 2D model. The sidewalls required for the 3D model were set to free slip, following Favier et al. (2012). To minimise the effect of the sidewalls on the boundary conditions, the ice velocity was measured at the center of the width (i.e. 5 km) (see the red dot in the figure below). To reduce the computational cost of 3D modelling, we developed a 3D model 8000 km long and 10 km wide around the initial grounding line. The mesh size along the Z axis is kept the same as in the 2D models (i.e., 0.9 km). Please note that the mesh-size in the in-plane direction (Z axis) is very large (i.e., 1 km) to save computational cost. However, the time-step size (0.01 year) of the 2D model is maintained in the 3D model.

[Figure]

Model setup in 3D idealised slope model

The modelling results are shown below:

[Figure]

Fig. S6 in the revised manuscript

Figures a-k show the ice velocity distribution at time = 100 years for A values ranging from $10^{-35}$ to $10^{-29}$ Pa$^{-4}$s$^{-1}$. We found that the overall velocity changes are very similar according to the A values (Figure m). The range of constrained A values was also derived as $10^{-35}$ to $2.5 \times 10^{-30}$ Pa$^{-4}$s$^{-1}$, as in the 2D model.

*There should be a demonstration of the mesh independence of the results, especially considering the saw-tooth oscillations in Fig 2., for both the idealized and realistic cases. This can just be included as a supplementary figure.*

**Authors' reply:**

Following the reviewer's suggestion, we have included the mesh size test (i.e., resolution test) in the Supplementary Material of the revised manuscript. We believe that the resolution test is a critical for assessing the effect of the mesh size on the overall results of our study. In particular, minimising the effect of mesh size is important for determining the range of A values, which is the main objective of our study.

Therefore, we tested three different mesh sizes, *i*) the mesh size used in the original draft (900 m), *ii*) a half of the original mesh size (450 m), *iii*) a double of the original mesh size (1800 m), and *iv*) a quadruple the original mesh size (3600 m). Although the resolution tests were performed for all models (including the idealised slope and Antarctic bed topography models) and all A values, we have included the nine cases (i.e., Ross, Ronne, Cook, Thwaites, Amery, Shackleton Ice Shelf, Mertz Glacier Tongue, Pine Island Glacier and the idealised slope case) with $A = 4.0 \times 10^{-32} \, \text{Pa}^{-4}\text{s}^{-1}$ in the Supplementary Material. Please see the figures below.

[Figure]

The resolution tests with different mesh sizes showed that the velocities at 100 years are similar as the mesh size decreases. For example, in the idealised slope model (see figure a above), the resolution tests with all mesh sizes of 450 m, 900 m, 1800 m and 3600 m gave velocities of 93 m/year. A similar trend was found for the Antarctic bed topography models (see figures b-i).

For most Antarctic ice shelves, the 3600 m mesh size models have a significantly different ice velocity at 100 years compared to the 900 m mesh size models. However, the 450 m mesh size models show similar ice velocities to the 900 m mesh size models. For example, the ice velocity of the Rone Ice Shelf is similar for the 900 m and 450 m mesh sizes. For Pine Island Glacier, the velocity is 210 m for the 3600 m mesh size. However, the ice velocities of all other models in 1800, 900 and 450 m mesh sizes converge to 120 m/year. Ice velocities tend to be similar at mesh sizes of ~900 m. This mesh independence of velocity confirms the suitability of the 900 m mesh size chosen for our study.

For the idealised models, we found that an increase in mesh size correlated with longer oscillation periods of velocities and grounding line position. This indicates that the influence of the mesh size on the grounding line motion in the idealised slope model is significant. The occurrence of this oscillation across all mesh sizes confirms that the physical motion is independent of mesh size. In the original draft, we had already discussed that the oscillations were due to the discrete nature of the grounding line position definition. To quote a sentence from the original manuscript:
* * *
**In the Results section of the original manuscript,**

The oscillation in ice velocity in Figure 2a is caused by the interaction with the discrete variations in the position of the grounding line, which is determined by the mesh size. As the grounding line advances, the ice velocity decreases rapidly due to the increase in frictional stress. This dynamics leads to oscillations in the overall ice velocity (see Fig. S2 in the Supporting Information).
* * *
The oscillations were not pronounced for mesh sizes of 450 m, 900 m, 1800 m, and 3600 m. For the Antarctic bed topography model, the oscillation behaviour with the mesh size showed a distinct pattern compared to the idealised slope model. In particular, the oscillations were not pronounced for mesh sizes of 450 m, 900 m and 1800 m. This discrepancy can be attributed to the complex bed topography and lower surface interface (e.g., tills and cavities) in a realistic setting. In addition, the complex bed topography of Antarctica results in a more gradual movement of the grounding line, as opposed to the abrupt changes in the idealised slope model. However, when the mesh size was too large (e.g., 3600 m), oscillations in the position and velocity of the grounding line occurred, highlighting the importance of selecting a sufficiently small mesh size.

*The definition in Eq. 6 and your values of m = 3,5,7,9 later are inconsistent: with these m values Eq. 6 should read 1/m not m.*

**Authors' reply:**

We greatly appreciate the reviewer's careful review of our manuscript, especially the inconsistency between Equation 6 and the chosen values of m. We have thoroughly checked the formulae and parameters for the Weertman friction law using the relevant literature. Please refer to the box below.

[Figure]

Brondex et al., 2017

We acknowledge that the correct presentation does indeed require expressing m as 1/m in order to align our study with the established theoretical and empirical frameworks documented in the literature. This correction is particularly important given the influential contributions of Brondex et al. (2017), Barnes and Gudmundsson (2022) to our understanding of the Weertman friction law. We have revised Equation 6 in our manuscript with 1/m to ensure that our approach is consistent with both of our data set for m = 3, 5, 7, and 9. We sincerely appreciate the reviewer's guidance in identifying this critical error.

**Table 1.** Values and Descriptions of parameters

| Description | Symbols | Values [unit] and equations |
|---|---|---|
| ⋮ | ⋮ | ⋮ |
| Fluidity parameter | $A$ | $10^{-35}$ to $10^{-29}$ [$Pa^{-4} \cdot s^{-1}$] |
| Friction coefficient | $C_w$ | $7.624 \times 10^6$ [$Pa \cdot m^{-1/3} \cdot s^{1/3}$] |
| Sliding exponent | $m$ |  1/3, 1/5, 1/7, and 1/9 [-] |
| ⋮ | ⋮ | ⋮ |

*Technical Comments*

*Line 30-31: By incorporating geodetic creep data… I'm not sure what is meant precisely by this phrase. Furthermore, while the Cuffy and Behn do support n=3, it would be relevant to say here that n=3 has been used since Glen's classic experiments (Glen, 1952).*

**Authors' reply:**

We thank the reviewer for clarifying the phrase "incorporation of geodetic creep data" in our manuscript. We are referring to the integration of measurements from different geodetic techniques, such as InSAR, with numerical modelling. By integrating geodetic creep data with ice sheet modelling, this method can provide insight into ice flow velocity and deformation, which is essential for improving the accuracy and reliability of the ice sheet model.
* * *
**In the Introduction section,**

Glen's experiments contributed significantly to the understanding of ice deformation, revealed the complex relationship between stress and strain rate (Glen, 1952; Glen, 1958). In ice sheet modelling, the value of $n$ of approximately 3 has been used, which effectively reproduces surface ice velocities in Antarctica (Martin and Sanderson, 1980; Pattyn et al., 2012), such as the Amery Ice Shelf (Thomas, 1973; Hamley et al., 1985).
* * *
*Throughout the manuscript mathematical variables in the text should be written in math mode, i.e. A and n = 4 rather than A and n=4.*

**Authors' reply:**

We have carefully revised the document to ensure that all mathematical variables are formatted in LaTeX math mode (e.g., $A$ and $n = 4$) for the readability of the manuscript. This correction has been applied throughout the manuscript. Please let us show the reviewer the examples of the correction.
* * *
**In the Abstract section,**

···The suggested range of the fluidity parameter ( $A$) for $n = 4$ is of the order of six (i.e., $10^{-35}$ to $10^{-29}$ Pa$^{-4}$s$^{-1}$), leading to a significant uncertainty in ice velocity than when $n = 3$.

**In Introduction section,**

···The value of  $A$ derived from a laboratory ice deformation experiment with $n = 3$ ranged from 1.8 to $93 \times 10^{-25}$ Pa$^{-3}$s$^{-1}$ (MacAyeal et al., 1998),

**In Method section,**

···when  $n = 4$, comparing ice flow velocity over a total period of 100 years.

> **In Results section,**
> ⋯fluidity parameter ( $A$) when  $n = 4$ (power-law stress exponent).
> **In the Discussion section,**
> instead of  $n = 3$, better explains ice flow in Antarctica (e.g., Behn et al., 2021; Ranganathan et al., 2021).
> **In the Conclusions section,**
> ⋯on ice flow velocity, with  $n = 4$ ⋯

*Line 85: should be lowercase p for pressure*

**Authors' reply:**

We thank the reviewer for bringing this error to our attention and apologise for any confusion it may have caused.

> **In the Method section,**
> $\eta$, $\rho_i$, **g**, and  $p$ indicate the ice viscosity, ice density, gravitational acceleration, and pressure, respectively.

*Around Eq. 4 and 5 $\varepsilon_e$ should be defined or the table should be referenced*

**Authors' reply:**

In response to this comment, we have added a reference to the Table 1, where the effective strain rate ($\dot{\varepsilon}_e$) is defined. Specifically, we have placed the sentence near equations 4 and 5, directing the readers to Table 1 for the definition of $\dot{\varepsilon}_e$.

> **In the Method section,**
>
> ⋯Glen's Law (Eq. 5) determines the viscosity as a function of A value and the power of the effective strain rate $\dot{\varepsilon}_e$ (see Table 1), representing the magnitude of the strain rate tensor.
>
> $$\eta = 2^{-1}A^{-1/n}\dot{\varepsilon}_e^{(1-n)/n}$$
> (5)
>
> **Tabel 1**
>
> | Strain rate | $\dot{\varepsilon}_{xx}$, $\dot{\varepsilon}_{yy}$ | [1/s] |
> |---|---|---|
> | In-plane strain rate | $\dot{\varepsilon}_{zz}$ | $-\left(\dot{\varepsilon}_{xx} + \dot{\varepsilon}_{yy}\right)$ [1/s] |
> | Effective strain rate | $\dot{\varepsilon}_e$ | $\left(0.5\left(\dot{\varepsilon}_{xx}^2 + 2\dot{\varepsilon}_{xy}^2 + \dot{\varepsilon}_{yy}^2 + \dot{\varepsilon}_{zz}^2\right)\right)^{0.5}$ [1/s] |

*The range of Fluidity parameters in Fig 2 should be expressed in exponent notation (i.e. 10^-29 rather than 0.001 x 10^-32)*

**Authors' reply:**

Regarding the reviewer's comments on the presentation of the range of A values in Figure 2, we appreciate the suggestion to change the exponential notation for accuracy. The recommendation to present these values in a more standardised form, such as $10^{-29}$ rather than $0.001 \times 10^{-32} \mathrm{Pa}^{-4}\mathrm{s}^{-1}$, will be incorporated in the revised manuscript. An excerpt of the manuscript and figure with the change in exponential notation is shown below.

[Figure]

Original Figure 2          Revised Figure 2.

*Line 150 of the text references dotted lines showing the positing of BedMap2 grounding lines in Fig 3, yet I cannot see them in this figure.*

**Authors' reply:**

We appreciate the reviewer's attention to detail in evaluating our manuscript. We acknowledge the confusion caused by the absence of a dotted line to indicate the location of the BedMap2 grounding line in Figure 3. To rectify this, we have revised Figure 3 to ensure that the dotted lines clearly represent the BedMap2 grounding lines as originally intended.

[Figure]

Original Figure 3.

Revised Figure 3.

We thank Reviewer once again for the valuable time and consideration.

Sincerely,

Sujeong Lim and Prof. Byung-Dal So

**References for this response letter**

Adams, C. J. C., Iverson, N. R., Helanow, C., Zoet, L. K., and Bate, C. E.: Softening of temperate ice by interstitial water, Front. Earth Sci., 9, 702761, https://doi.org/10.3389/feart.2021.702761, 2021.

Azuma, N.: A flow law for anisotropic ice and its application to ice sheets, Earth Planet. Sc. Lett., 128, 601–614, https://doi.org/10.1016/0012-821X(94)90173-2, 1994.

Barnes, J. M. and Gudmundsson, G. H.: The predictive power of ice sheet models and the regional sensitivity of ice loss to basal sliding parameterisations: a case study of Pine Island and Thwaites glaciers, West Antarctica, The Cryosphere, 16, 4291–4304, https://doi.org/10.5194/tc-16-4291-2022, 2022.

Behn, M. D., Goldsby, D. L., and Hirth, G.: The role of grain size evolution in the rheology of ice: implications for reconciling laboratory creep data and the Glen flow law, The Cryosphere, 15, 4589–4605, https://doi.org/10.5194/tc-15-4589-2021, 2021.

Brondex, J., Gagliardini, O., Gillet-Chaulet, F., and Durand, G.: Sensitivity of grounding line dynamics to the choice of the friction law, J. Glaciol., 63, 854–866, https://doi.org/10.1017/jog.2017.51, 2017.

Brown, J., Harper, J., and Humphrey, N.: Liquid water content in ice estimated through a full-depth ground radar profile and borehole measurements in western Greenland, The Cryosphere, 11, 669–679, https://doi.org/10.5194/tc-11-669-2017, 2017.

Cuffey, K. M. and Paterson, W. S. B.: The physics of glaciers, Academic Press, Elsevier, Burlington, ISBN 9780123694614, 2010.

Duddu, R. and Waisman, H.: A temperature dependent creep damage model for polycrystalline ice, Mech. Mater., 46, 23–41, 2012.

Favier, L., Gagliardini, O., Durand, G., and Zwinger, T.: A three dimensional full Stokes model of the grounding line dynamics: effect of a pinning point beneath the ice shelf, The Cryosphere, 6, 101–112, http://doi.org/10.5194/tc-6-101-2012, 2012.

Gillet-Chaulet, F., Gagliardini, O., Meyssonnier, J., Zwinger, T., and Ruokolainen, J.: Flow-induced anisotropy in polar ice and related ice-sheet flow modelling, J. Non-Newton. Fluid, 134, 33–43, https://doi.org/10.1016/j.jnnfm.2005.11.005, 2006.

Goldsby, D. and Kohlstedt, D.: Superplastic deformation of ice: Experimental observations, J. Geophys. Res.-Solid Earth, 106, 11017–11030, https://doi.org/10.1029/2000JB900336, 2001.

Hruby, K., Gerbi, C., Koons, P., Campbell, S., Martín, C., and Hawley, R.: The impact of temperature and crystal orientation fabric on the dynamics of mountain glaciers and ice streams, J. Glaciol., 66, 755–765, http://doi,org/10.1017/jog.2020.44, 2020.

Hudleston, P. J.: Structures and fabrics in glacial ice: A review, J. Struct. Geol., 81, 1–27, https://doi.org/10.1016/j.jsg.2015.09.003, 2015.

Jansen, D., Sandhäger, H., and Rack, W.: Model experiments on large tabular iceberg evolution: ablation and strain thinning, J. Glaciol., 51, 363–372,

http://doi.org/10.3189/172756505781829313, 2005.

Lilien, D. A., Rathmann, N. M., Hvidberg, C. S., and Dahl-Jensen, D.: Modeling Ice-Crystal Fabric as a Proxy for Ice-Stream Stability, J. Geophys. Res.-Earth, 126, e2021JF006306, https://doi.org/10.1029/2021JF006306, 2021.

Millstein, J. D., Minchew, B. M., and Pegler, S. S.: Ice viscosity is more sensitive to stress than commonly assumed, Communications Earth & Environment, 3, 57, https://doi.org/10.1038/s43247-022-00385-x, 2022.

Stone, B.M., Jordaan, I.J., Xiao, J. and Jones, S.J.: Experiments on the damage process in ice under compressive states of stress, J. Glaciol., 43(143), 11-25, https://doi.org/10.3189/S002214300000277X, 1997.

Wilson, C. J. L. and Peternell, M.: Ice deformed in compression and simple shear: control of temperature and initial fabric, J. Glaciol., 58, 11–22, http://doi.org/10.3189/2012JoG11J065, 2012.